# DNAJA2 deficiency activates cGAS-STING pathway via the induction of aberrant mitosis and chromosome instability

Yaping Huang[1,6], Changzheng Lu [2,3,6], Hanzhi Wang[1], Liya Gu[1], Yang-Xin Fu[2,4] ✉ & Guo-Min Li [1,5] ✉

Molecular chaperone HSP70s are attractive targets for cancer therapy, but their substrate broadness and functional non-specificity have limited their role in therapeutical success. Functioning as HSP70's cochaperones, HSP40s determine the client specificity of HSP70s, and could be better targets for cancer therapy. Here we show that tumors defective in HSP40 member DNAJA2 are benefitted from immune-checkpoint blockade (ICB) therapy. Mechanistically, DNAJA2 maintains centrosome homeostasis by timely degrading key centriolar satellite proteins PCM1 and CEP290 via HSC70 chaperone-mediated autophagy (CMA). Tumor cells depleted of DNAJA2 or CMA factor LAMP2A exhibit elevated levels of centriolar satellite proteins, which causes aberrant mitosis characterized by abnormal spindles, chromosome missegregation and micronuclei formation. This activates the cGAS-STING pathway to enhance ICB therapy response in tumors derived from DNAJA2-deficient cells. Our study reveals a role for DNAJA2 to regulate mitotic division and chromosome stability and suggests DNAJA2 as a potential target to enhance cancer immunotherapy, thereby providing strategies to advance HSPs-based cancer therapy.

The DnaJ heat shock protein (HSP40) family member A2 (DNAJA2) is one of the most abundant DNAJ cochaperones of HSP70 or HSC70 (the 71 kDa heat shock cognate)[1–4]. These HSPs play important roles in diverse biological pathways by maintaining protein homeostasis, such as protein folding, refolding, assembly, and protein degradation via proteolysis pathways including chaperone-mediated autophagy (CMA)[2,3]. Dysregulations of HSPs predispose to human diseases including cancer[3,5,6]. For instance, intracellular HSP70 can block the function of key factors involved in the apoptosis and autophagy machineries[7]. In cancer cells, overexpression of HSP70 promotes tumor growth, metastasis, and resistance to chemotherapy[5,6]. Conversely, depletion of HSP70 in animal models inhibits tumor growth[7]. Extracellular HSP70 functions to activate anti-tumor immunity[8]. Therefore, tremendous efforts have been made in targeting HSP70 for

cancer therapy in the past[9–11]. However, because of functional non-specificity and global effects of HSP70, these efforts do not translate into clinical success[5,6,9,10].

As cochaperones of HSP70s, HSP40 proteins determine the specificity and diversity of HSP70's clients and functions by selectively targeting different client proteins[4,12,13], thereby dramatically increasing the selectivity and specificity of HSP70 networks in regulating cellular pathways. Therefore, HSP40s are likely better targets for cancer treatment. DNAJA2, an important member of the DNAJA subfamily of HSP40s, is involved in chaperone HSC70-mediated protein homeostasis[14–16]. Interestingly, the yeast homolog of DNAJA2, Ydj1, has been shown to play important roles in regulating cell cycle[17] and DNA damage response[18,19]. It is possible that DNAJA2 also plays similar roles in regulating cell division and DNA repair. Thus, defects in DNAJA2 may

[1]Department of Radiation Oncology, University of Texas Southwestern Medical Center, Dallas, TX, USA. [2]Department of Pathology, University of Texas Southwestern Medical Center, Dallas, TX, USA. [3]Institute of Cancer Research, Shenzhen Bay Laboratory, Shenzhen, China. [4]Department of Basic Medical Sciences, Tsinghua University School of Medicine, Beijing, China. [5]Chinese Institutes for Medical Research, Beijing, China. [6]These authors contributed equally: Yaping Huang, Changzheng Lu. ✉e-mail: Yang-Xin.Fu@utsouthwestern.edu; Guo-Min.Li@UTSouthwestern.edu

cause genomic instability, which can drive cancer development and influence cancer treatment[20,21]. Recent studies have shown that genomic instability in cancer benefits cancer immunotherapy, as it determines the immunogenicity of tumor cells through mechanisms including activating the cytosolic innate immunity mediated mainly by the cGAS-STING pathway[22].

Cells have evolved multiple mechanisms to ensure genome integrity throughout the cell division cycle, which can be divided into the interphase (G1, S, and G2) and cell division or mitotic division (M), where chromosome segregation and all other cell component separation into two daughter cells occur. While DNA damage removal on DNA molecules by various DNA repair pathways during interphase is important, maintaining accurate chromosome segregation and chromosome number during mitotic division is absolutely critical for genome integrity, as aberrant mitotic division causes cancer and other diseases[23–25]. Given the important role that HSP40 proteins play in protein homeostasis, we hypothesize that DNAJA2 ensures genome stability by regulating cell division and/or DNA repair pathways, and its deficiency modulates genomic instability and cancer immunotherapy.

Here we show that DNAJA2 colocalizes with HSC70 at centrosomes and is required for timely degradation of centriolar satellite proteins PCM1 and CEP290 via the HSC70-mediated CMA pathway, thereby maintaining centrosome homeostasis and mitotic integrity. Tumor cells defective in DNAJA2, HSC70, or CMA factor LAMP2A exhibit abnormal spindles and chromosome segregation errors, which induce aneuploidy and micronuclei. The formation of micronuclei activates the cGAS-STING-mediated type I interferon pathway. Consequently, tumors derived from DNAJA2-deficient cancer cells in animal models are highly responsive to the immune checkpoint blockade (ICB) therapy, which correlates with the ICB therapy result observed in cancer patients. Therefore, this study has not only established a role of the HSP40-CMA axis in maintaining mitotic integrity, but also identified DNAJA2 and the CMA pathway factors as potential targets to enhance cancer immunotherapy, providing strategies to advance HSPs-based cancer therapy.

## Results

### DNAJA2 deficiency leads to aberrant mitosis
To explore the role of DNAJA2 in maintaining genome stability, particularly in maintaining mitotic integrity as DNAJA2 has been identified in the centrosome proteome[26,27], we knocked out DNAJA2 (DJ2$^{-/-}$) in HeLa cells and monitored their mitotic division. Interestingly, we found that DNAJA2 knockout significantly elevated the production of multinuclear cells, and restoration of DNAJA2 expression in DJ2$^{-/-}$ cells restored the multinuclear cell percentage to the normal level (Fig. 1a, b and Supplementary Movie 1). These results suggest that DNAJA2 is an important regulator of genome integrity during mitosis. To further determine mitotic abnormalities in DJ2$^{-/-}$ cells, we ectopically expressed EGFP-tagged histone H2B (EGFP-H2B) in WT and DJ2$^{-/-}$ HeLa cells, and performed time-lapse live cell imaging analyses. As shown in Fig. 1c–e, DNAJA2-depleted cells showed delayed mitotic exit and increased rate of chromosome segregation errors (Supplementary Movie 2). Taken together, DNAJA2 is an important regulator of mitotic division, and its defects lead to aneuploidy and defective mitosis.

### DNAJA2 defects cause abnormal mitotic spindles and chromosome alignments
To determine the mechanism by which DNAJA2 regulates mitosis, we ectopically expressed EGFP-H2B and mCherry-tagged α-tubulin (mCherry-αtubulin) in WT and DJ2$^{-/-}$ HeLa cells, and performed live cell imaging analyses. At the onset of mitosis, the majority of DNAJA2-depleted cells showed two spindle poles similar to WT cells (Fig. 1f and Supplementary Movie 3). However, in DNAJA2-depleted cells, the two spindle poles undergo rapid fragmentation to form diffused or multipolar spindles (Fig. 1f and Supplementary Movie 3), suggesting that DNAJA2 defects cause mitotic spindle pole fragmentation and abnormal mitotic spindles. This was confirmed after we analyzed prophase cells, which showed intact bipolar centrosomes in both WT and DJ2$^{-/-}$ HeLa cells (Supplementary Fig. 1a, b). The quantification data of mitotic cells showed that most of the cells exhibiting spindle pole fragmentation and abnormal spindles underwent multipolar-spindle division or displayed chromosome segregation errors even though the abnormal spindles were corrected to form a pseudo-bipolar spindle before the onset of anaphase (Fig. 1g). These results strongly argue that the generation of elevated multinuclear cells and chromosome segregation errors in DNAJA2-deficient cells results from mitotic spindle pole fragmentation and aberrant spindle formation.

The morphology of mitotic spindles was analyzed in randomly-picked mitotic cells from DNAJA2-proficient and -deficient lines after fixation and staining for α-tubulin. While the vast majority of DNAJA2-proficient HeLa cells showed normal bipolar spindles, approximately 50% of the DNAJA2-deficient mitotic cells displayed abnormal spindles, including multiple-polar, mono-polar, and diffused ones (Fig. 1h, i). Similar results were also obtained in DNAJA2-deficient human retinal pigment epithelial-1 (RPE1) cells, lung cancer H460 cells, colon cancer SW620 cells, and DNAJA2-deficient mouse melanoma B16-OVA cells, as compared with their WT controls (Supplementary Fig. 1c–g). To test if the abnormal spindles cause chromosome misalignment and congressional defects, which eventually leads to segregation errors[28,29], we analyzed the individual phases of mitosis by immunofluorescence analysis using an anti-centromere antibody (ACA). As expected, ~80% of WT HeLa cells showed well-aligned and congressed chromosomes with centromeres clustered at the mid-plate, and only ~20% cells showed abnormality (Fig. 1j). However, ~55% of DNAJA2-depleted cells showed abnormal chromosome alignment and/or lagging chromosomes (Fig. 1j, k). These data suggest that depleting DNAJA2 causes spindle assembly defects, which further impair the downstream chromosome alignment and segregation.

### DNAJA2 regulates the turnover of key centriolar satellite proteins
As the organization center for mitotic spindles, centrosomes play a key role in ensuring spindle bipolarity[29]. The observed aberrant spindle assembly in DNAJA2-depleted cells prompted us to speculate that DNAJA2 regulates centrosome homeostasis. To test this possibility, we determined the subcellular localization of DNAJA2 by immunofluorescence analysis. The results showed that DNAJA2 colocalized with centrosome proteins PLK1 and NUDC (Fig. 2a), suggesting that DNAJA2 localized at centrosomes. We also found that DNAJA2 colocalized with HSC70 at centrosomes (Fig. 2b), consistent with the fact that DNAJA2 is a cochaperone of HSC70[2]. These observations suggest that the HSC70/DNAJA2 chaperone complex may regulate the homeostasis of centrosome proteins by mediating their degradation in a timely manner. We therefore measured protein levels of several important centrosome proteins, including PCM1, CEP290, CEP131, SSX2IP, Pericentrin, and Centrin. Interestingly, we observed significantly elevated levels of PCM1 and CEP290, but not others in DNAJA2-deficient mouse breast cancer cell line 4T1 and HeLa cells (Fig. 2c, d and Supplementary Fig. 2a–d). Consistently, the immunofluorescence results showed much stronger intensity of these two proteins at interphase in DNAJA2-depleted cells (Fig. 2e, f and Supplementary Fig. 2k), suggesting that this phenomenon is not specific to mitotic cells. Protein half-life analysis also showed that both PCM1 and CEP290 were more stable in DNAJA2-depleted cells than in WT cells (Fig. 2g–i), indicating that DNAJA2 regulates the turnover of key centriolar satellite proteins at centrosomes.

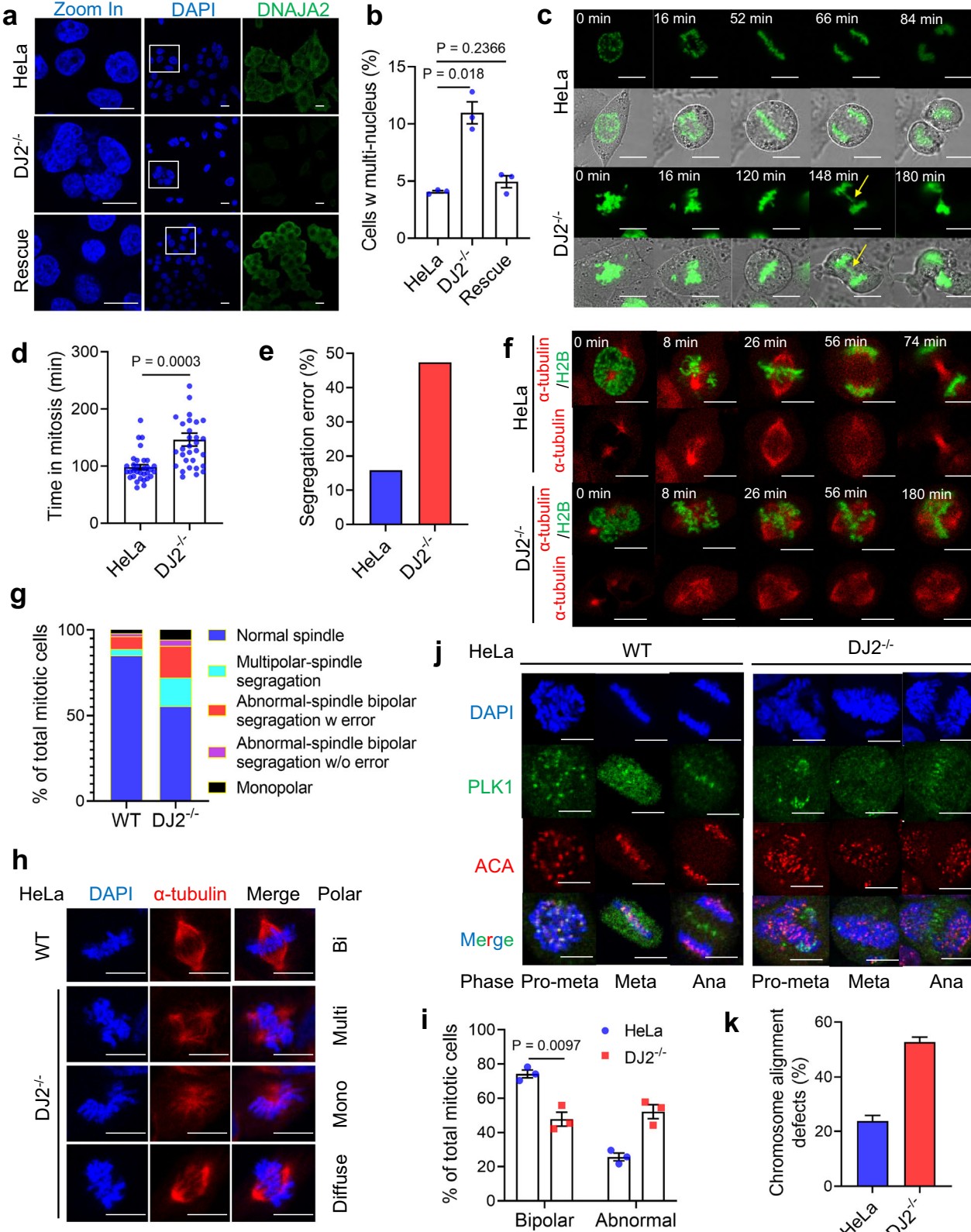

## DNAJA2 is essential for CMA-mediated degradation of PMC1 and CEP290

HSP40 proteins are known to promote protein degradation through the lysosomal pathway[2,14,15,30]. To determine if DNAJA2 is involved in degradation of PCM1, CEP290 via the lysosomal pathway, we first treated cells with autophagy inhibitor chloroquine (CQ). The results showed that CQ dramatically stabilized PCM1 and CEP290 (Fig. 3a, b

and Supplementary Fig. 2e, f), indicating that DNAJA2 promotes PCM1/ CEP290 degradation through the lysosomal pathway. We then measured the degradation rate of PCM1 and CEP290 in WT and DNAJA2-KO HeLa or 4T1 cells treated with lysosome protease inhibitors ammonium chloride and leupeptin (combined as NL). We found that the treatment resulted in significant increase of the protein levels of both PCM1 and CEP290 in WT, but not in DNAJA2-deficient cells (Fig. 3c, d

**Fig. 1 | DNAJA2 deficiency leads to aberrant mitosis. a, b** Representative images and quantification (means ± SEM, $n = 3$ experimental repeats) of multinucleate cells in control, $DJ2^{-/-}$ and $DJ2$-rescued HeLa cells. **c** Representative images showing time-lapsed mitotic cell division of control and $DJ2^{-/-}$ HeLa cells. Lagging chromosomes are indicated by yellow arrows. **d, e** Quantifications of average mitosis time (HeLa $n = 32$ cells, $DJ2^{-/-}$ $n = 31$ cells) (**d**) and chromosome segregation error rate (HeLa $n = 40$ cells, $DJ2^{-/-}$ $n = 48$ cells) (**e**) of control and $DJ2^{-/-}$ HeLa cells. **f** Representative images showing spindle morphologies (red) and chromosome segregation (green) in time-lapsed mitotic cell division of control and $DJ2^{-/-}$ HeLa cells. **g** Quantifications of mitotic phenotypes from multiple independent live imaging experiments, $n = 54$ cells. **h** Representative images showing spindle morphologies (red) of control and $DJ2^{-/-}$ HeLa cells. **i** Quantification of cells with bipolar and abnormal spindles in control and $DJ2^{-/-}$ HeLa cells (means ± SEM, $n = 3$ experimental repeats). All multi-polar, monopolar, and diffuse-polar spindles were counted as abnormal spindles. **j, k** Representative images and quantification of mitotic cells with normal and defective chromosome alignment in control and $DJ2^{-/-}$ HeLa cells (means ± SEM, $n = 2$ experimental repeats). The well-lined anti-centromere antibody (ACA) foci indicate normal alignment, and the diffused ACA foci suggest abnormal alignment. Scale bar, 20 μm. $P$ values were determined by two-tailed unpaired t test with Welch's correction. Source data are provided as a Source data file.

and Supplementary Fig. 2g–j). Consistently, the NL treatment significantly increased the colocalized foci between PCM1 and lysosome receptor LAMP2A in WT but not in DNAJA2-deficient cells (Supplementary Fig. 2k, l). These results suggest that DNAJA2 is essential for lysosomal degradation of PCM1 and CEP290.

Since HSP40 members serve as cochaperones of HSC70 in the CMA-mediated protein degradations[14,15], we examined if DNAJA2-promoted PCM1/CEP290 degradation is through the HSC70-mediated CMA pathway. We measured the protein level of PCM1/CEP290 after treating HeLa cells with 10 μM HSC70 inhibitor apoptozole (AZ). The results showed that the treatment stabilized PCM1 and CEP290 in a time-dependent manner (Fig. 3e, f), suggesting that HSC70 is indeed involved in degrading PCM1 and CEP290. To definitively address that CMA is responsible for PCM1/CEP290 degradation, we knocked out the CMA-specific lysosome receptor gene *LAMP2A* in both HeLa and 4T1 cells, and measured the protein levels of PCM1 and CEP290 in *LAMP2A* knockout ($L2A^{-/-}$) and control cells. As shown in Fig. 3g, h and Supplementary Fig. 2a, b, both PCM1 and CEP290 showed elevated levels in $L2A^{-/-}$ cells, as compared with WT controls. In addition, both proteins exhibited a longer half-life in $L2A^{-/-}$ cells than in control cells (Fig. 3i–k), as observed in DNAJA2-deficient cells (Fig. 2g–i). In conclusion, PCM1 and CEP290 are indeed degraded through the CMA pathway. Given that both PCM1 and CEP290 behaved the same in DNAJA2-facilitated degradation, and that PCM1 plays essential roles in maintaining centrosome homeostasis and mitotic integrity[31–33], we decided to focus on PCM1 to investigate the role of DNAJA2 in regulating mitotic integrity.

To determine whether DNAJA2 functions to promote PCM1 degradation in the CMA pathway, we measured PCM1 degradation in cells with or without DNAJA2 after treating cells with the CMA activator AR7. The results showed that AR7 was able to induce PCM1 degradation more efficiently in WT controls and *DNAJA2*-rescued cells than in *DNAJA2*-deficient cells (Fig. 3l, m and Supplementary Fig. 3a), but the AR7-stimulated PCM1 degradation was not observed in $L2A^{-/-}$ cells (Fig. 3l, m). Together, these results indicate that PCM1 is degraded via the CMA pathway in a DNAJA2- and LAMP2A-dependent manner.

## PCM1 is a bona-fide substrate of the CMA pathway

HSC70 is known to recognize and deliver its substrate protein to lysosome for degradation in the CMA pathway[30]. We therefore hypothesized that as a cochaperone of HSC70, DNAJA2 may facilitate the substrate recognition and binding by HSC70 in CMA-mediated protein degradation. To test this hypothesis, we performed co-immunoprecipitation (Co-IP) to determine physical interactions between HSC70 and PCM1 in *DNAJA2*-proficient and -deficient cells using an HSC70 antibody. The results showed that the HSC70 antibody efficiently pulled down PCM1 in WT, but not in $DJ2^{-/-}$ 4T1 cells (Fig. 4a), suggesting that DNAJA2 promotes the HSC70-PCM1 interaction to facilitate PMC1 degradation via the CMA pathway.

The DNAJA2-promoted interaction between HSC70 and PCM1 prompted us to postulate that PCM1 is a substrate of the CMA pathway. An essential characteristic being a CMA substrate is the presence of the KFERQ pentapeptide motif[30]. We indeed identified 5 canonical KFERQ-like motifs in human PCM1 (Supplementary Fig. 3b). To determine if these KFERQ-like motifs are involved in the interaction with HSC70 to facilitate PCM1 degradation by the CMA pathway, we created two HA-tagged PCM1 mutants by substituting RQ to AA in two most accessible motifs (PCM1-2AA) or all five motifs (PCM1-5AA) (Supplementary Fig. 3b). The resulting WT and mutant PCM1s were expressed in HeLa cells and examined for their ability to interact with HSC70 and DNAJA2 by Co-IP using an anti-HA antibody. The results showed that the amount of co-IPed WT PCM1 was 4-fold higher than that of PCM1-2AA (Fig. 4b); we also observed a similarly elevated level of DNAJA2 in the precipitate from cells expressing WT PCM1, as compared with cells expressing the mutant PCM1 (Fig. 4b). These results suggest that the putative KFERQ-like motifs are essential for HSC70 binding.

To determine the impact of the KFERQ motifs on PCM1 degradation, we measured the stability of WT and KFERQ-mutated HA-PCM1s. The results revealed that both PCM1-2AA and PCM1-5AA were more stable than WT PCM1 (Fig. 4c, d and Supplementary Fig. 3c, d), indicating that the putative KFERQ-like motifs are required for PCM1 degradation. However, the degradation of PCM1 was inhibited in *DNAJA2*-depleted cells (Fig. 4e). To further confirm that the KFERQ-like motifs are required for lysosomal degradation of PCM1, we measured the degradation speed of WT PCM1 and PCM1-2AA in cells treated with NL. The results showed that the mutant PCM1 was indeed degraded much slower than the WT one (Fig. 4f, g). Therefore, we conclude that PCM1 is a bona-fide substrate of the CMA pathway, and its degradation, which is essential for maintaining centrosome integrity[32], requires physical interactions with the HSC70/DNAJA2 chaperone complex.

Since PCM1 was also reported to be degraded by ubiquitination proteasome system (UPS) and macroautophagy[32,34–36], we speculated that certain post-translational modifications of PCM1 may contribute to PCM1 degradation by CMA pathway, as post-translational modifications in or outside the KFERQ motifs, such as phosphorylation and acetylation, modulate CMA substrate recognition[30,37]. PCM1 is phosphorylated at Ser110 and Ser372 (Supplementary Fig. 3b) by PLK1 and PLK4 in G2/M and G1 phases, respectively[38,39]. To determine if PLK1- and PLK4-catalyzed phosphorylation of Ser110 and Ser372, which are near the N-terminal KFERQ-like motif, promotes PCM1 degradation by CMA, we measured PCM1 degradation rates in lysosomal protease inhibitor NL-treated cells in the presence or absence of the PLK1 inhibitor Volasertib or PLK4 inhibitor Centrinone. The NL treatment dramatically elevated PCM1 level in control cells but not in cells inhibited of the PLK1 or PLK4 kinase activity (Fig. 4h). Quantification data showed that PCM1 was no longer degraded through lysosomal pathway when PLK1 or PLK4 kinase was inhibited (Fig. 4i). Taken together, the phosphorylated PCM1 is targeted for lysosomal degradation by DNAJA2/HSC70-mediated CMA pathway.

## Persistent PCM1 contributes to the mitotic defects in DNAJA2-deficient cells

To determine if the persistent PCM1 level is responsible for the observed mitotic defects in DNAJA2-deficient cells, we partially knocked down *PCM1* in $DJ2^{-/-}$ HeLa cells to the level of HeLa cells (Fig. 5a), and measured micronuclei formation and mitotic phenotypes. The results showed that *PCM1* knockdown cells (shPCM1)

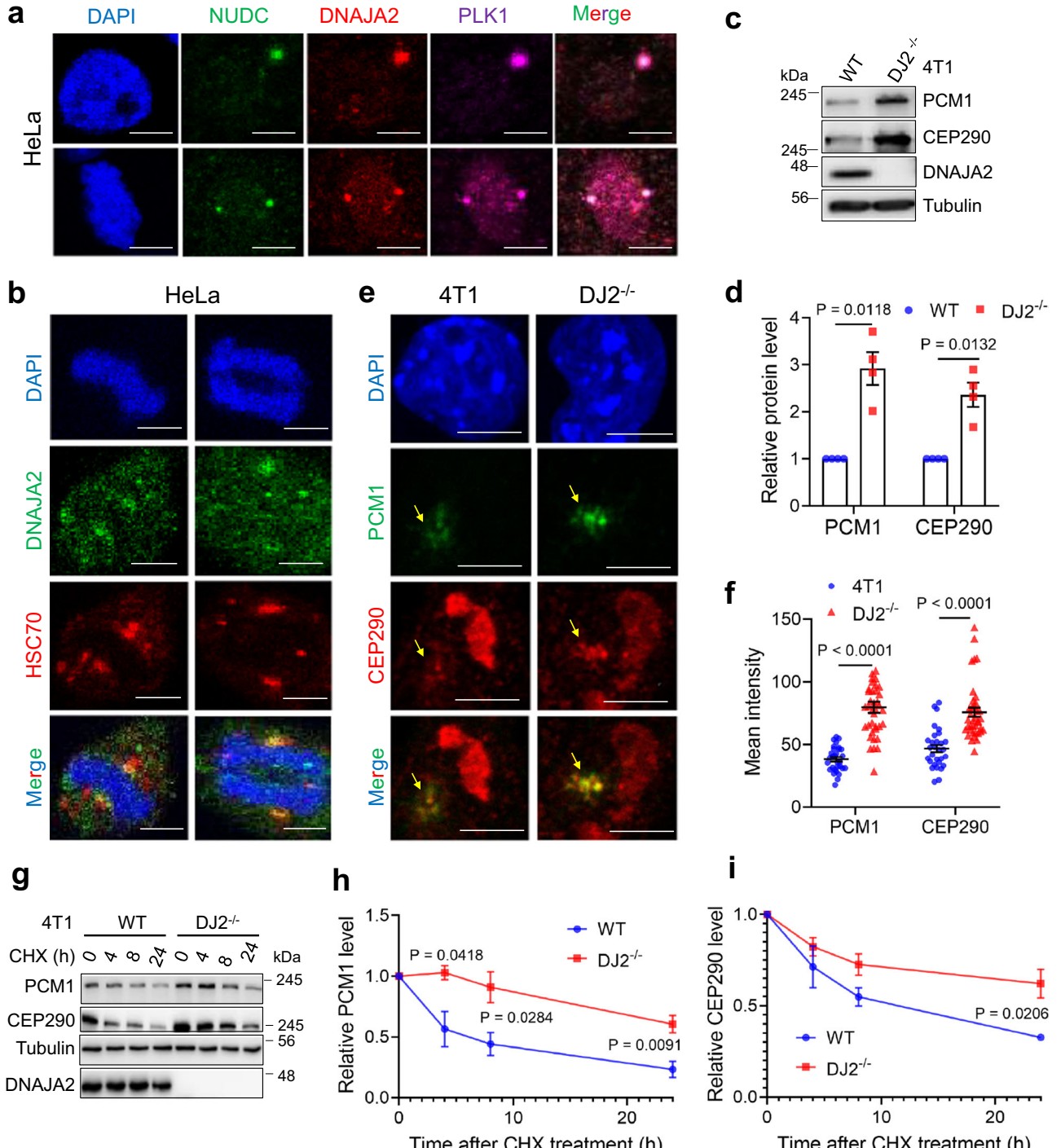

**Fig. 2 | DNAJA2 regulates the stability of key centriolar satellite proteins.**
**a** Representative images showing colocalization between DNAJA2 and centrosome proteins NUDC and PLK1 in both interphase (top) and M-phase (bottom) HeLa cells. **b** Confocal images showing colocalization between DNAJA2 and HSC70 at centrosomes of M-phase HeLa cells. Data in (**a**) and (**b**) were representative results from three independent experiments with similar outcomes. **c** Western blots showing protein levels of PCM1 and CEP290 in WT and *DJ2*−/− 4T1 cells. **d** Quantifications of relative PCM1 and CEP290 levels (means ± SEM, *n* = 4 experimental repeats) in WT and *DJ2*−/− 4T1 cells, as shown in (**c**). **e** Confocal images showing PCM1 and CEP290 signals (indicated by yellow arrows) in control and *DJ2*−/− 4T1 cells.

**f** Quantification (means ± SEM) of PCM1 and CEP290 intensities in control (*n* = 30 cells) and *DJ2*−/− (*n* = 38 cells) 4T1 cells, as shown in (**e**). **g** Western blots showing protein stability of PCM1 and CEP290 in WT and *DJ2*−/− 4T1 cells. Cells were treated with 50 µg/ml cycloheximide (CHX) for the indicated times and harvested for Western blotting analysis. **h, i** Quantifications (means ± SEM) of relative protein level of PCM1 (*n* = 4 experimental repeats) and CEP290 (*n* = 3 experimental repeats) as shown in (**g**). The relative protein levels were normalized to the level of the time 0 point in both groups. Scale bar, 10 µm. The *P* value in figure (**i**) was determined by two-tailed unpaired t test and the others were determined by two-tailed unpaired t test with Welch's correction. Source data are provided as a Source data file.

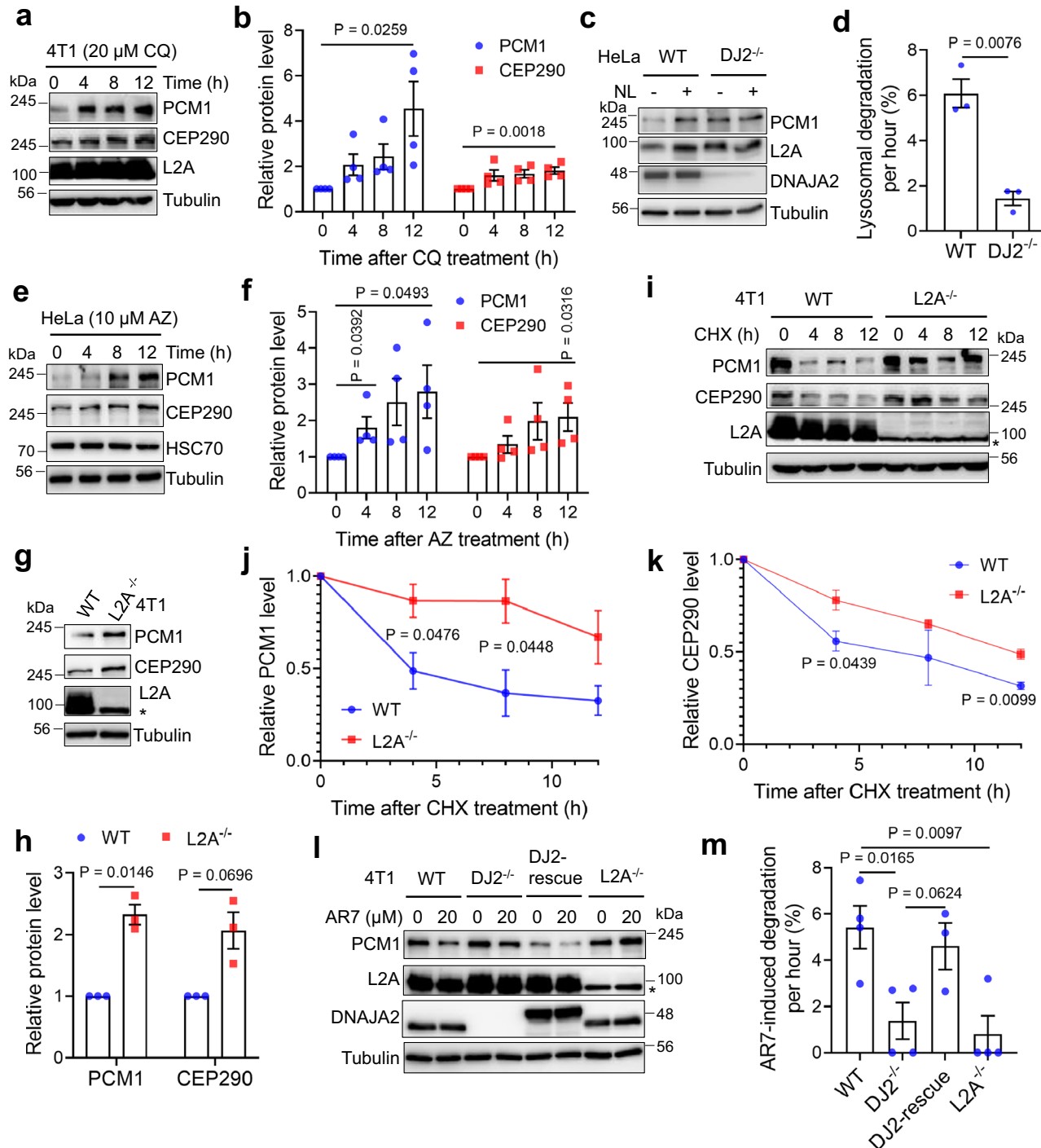

**Fig. 3 | DNAJA2 regulates PCM1/CEP290 degradation via chaperone-mediated autophagy. a** Western blotting analysis showing PCM1 and CEP290 protein levels in 4T1 cells treated with 20 μM chloroquine (CQ) for the indicated times. **b** Quantifications of relative protein levels (*n* = 4) of PCM1 and CEP290 as shown in (**a**). **c** Western blotting analysis showing PCM1 protein level in WT and *DJ2*⁻/⁻ HeLa cells treated with or without NL (20 mM ammonium chloride and 100 μM leupeptin) for 16 h. **d** Calculation of PCM1 degradation rate (*n* = 3) in lysosomes in WT and *DJ2*⁻/⁻ HeLa cells as shown in (**c**). Lysosomal degradation rate was calculated as: degradation per hour (%) = 100%(Relative PCM1 level of NL-treated group/Relative PCM1 level of NL-untreated group - 1)/16. **e** Western blotting analysis showing PCM1 and CEP290 protein levels in HeLa cells treated with 10 μM Apoptozole (AZ) for the indicated times. **f** Quantifications of relative protein levels (*n* = 4) of PCM1 and CEP290 as shown in (**e**). **g** Western blotting analysis showing the expression levels

of PCM1 and CEP290 in WT and *L2A*⁻/⁻ 4T1 cells. * indicates a non-specific band. **h** Quantifications of relative PCM1 and CEP290 levels (*n* = 3) in WT and *L2A*⁻/⁻ 4T1 cells, as shown in (**g**). **i** Protein stability analysis of PCM1 and CEP290 in WT and *L2A*⁻/⁻ 4T1 cells treated with 50 μg/ml cycloheximide (CHX), as indicated. **j, k** Quantifications of relative protein levels (*n* = 3) of PCM1 and CEP290 as shown in (**i**). **l** Western blotting analysis of PCM1 degradation in response to AR7 (a CMA activator) in WT, *DJ2*⁻/⁻, *DJ2*-rescued and *L2A*⁻/⁻ 4T1 cells. **m** Calculation of PCM1 degradation rate (*n* = 4) in response to AR7 treatment in WT, *DJ2*⁻/⁻, *DJ2*-rescued and *L2A*⁻/⁻ 4T1 cells as shown in (**l**). Data are shown as means ± SEM of *n* experimental repeats. The *P* values in figures (**b**), (**f**) were determined by two-tailed unpaired t test and the others were determined by two-tailed unpaired t test with Welch's correction. Source data are provided as a Source data file.

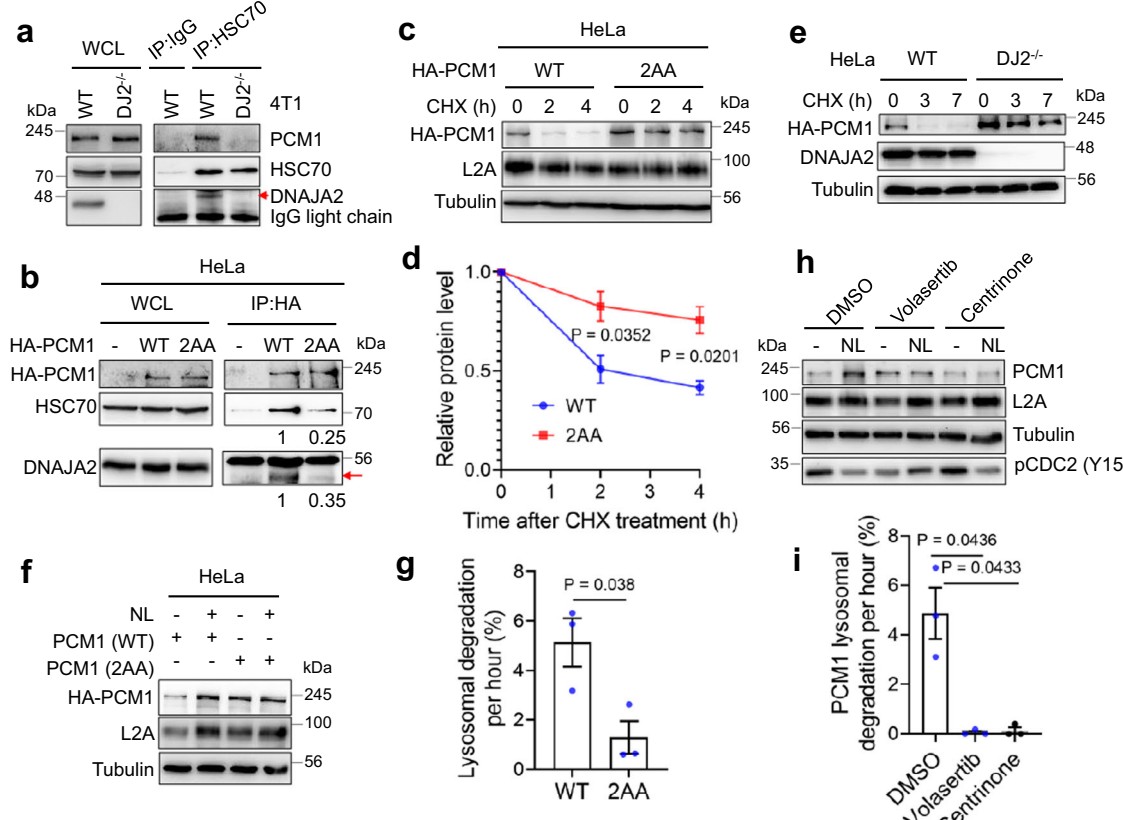

**Fig. 4 | PCM1 is a bona-fide substrate of CMA. a** Co-IP Western analysis showing co-immunoprecipitation of PCM1 by a HSC70 antibody in WT but not in *DJ2⁻/⁻* 4T1 cells, *n* = 3 independent experiments. **b** Co-IP Western analysis showing the importance of the PCM1 KFERQ-like motifs in physical interactions between HSC70, DNAJA2, and PCM1, *n* = 2 independent experiments. HA-tagged WT or PCM1 KFERQ mutant (2AA) was expressed in HeLa cells and the co-IP assays were performed using an anti-HA antibody. The co-IPed DNAJA2 band is indicated by a red arrow. **c** Western blots showing that HA-PCM1 degradation depends on its KFERQ-like motifs in HeLa cells. **d** Quantifications of relative protein levels (means ± SEM, *n* = 3 experimental repeats) for WT and mutant PCM1 as shown in (**c**). The relative protein level was compared with the amount of the protein in time 0 after normalization with the loading control, tubulin. **e** Western blots showing HA-PCM1 stability in WT and *DJ2⁻/⁻* HeLa cells, *n* = 2 independent experiments. **f** Western

blotting analysis showing that HA-PCM1 lysosomal degradation depends on its KFERQ-like motifs in HeLa cells. HeLa cells expressing WT or KFERQ mutant (2AA) HA-PCM1 were treated with or without NL (20 mM ammonium chloride and 100 μM leupeptin) for 16 h before harvesting for analysis. **g** Calculation of HA-PCM1 degradation rate (means ± SEM, *n* = 3 experimental repeats) in lysosomes as shown in (**f**). **h** Western blots showing that lysosomal degradation of PCM1 depends on the kinase activities of PLK1 and PLK4. HeLa cells in the presence or absence of a PLK1 kinase inhibitor Volasertib (50 nM) or a PLK4 inhibitor Centrinone (50 nM) were treated with NL for 16 h before harvesting for western blotting analysis. **i** Calculation of PCM1 degradation rate (means ± SEM, *n* = 3 experimental repeats) in lysosomes in HeLa cells treated with or without Volasertib or Centrinone as shown in (**h**). *P* values were determined by two-tailed unpaired t test with Welch's correction. Source data are provided as a Source data file.

exhibited decreased level of micronuclei as compared to the control knockdown (shCtrl) cells (Fig. 5b, c). Analysis of spindle morphology in fixed mitotic cells revealed that PCM1 knockdown significantly reduced the spindle abnormality in DNAJA2-deficient cells (Fig. 5d, e). In contrast, overexpression of PCM1, especially the CMA-inaccessible PCM1 mutant (PCM1-2AA), dramatically induces micronuclei (MN) formation in WT HeLa cells as compared to control cells expressing empty vector (Fig. 5f, g). These results suggest that excess amount of PCM1 is sufficient to drive mitotic defects. In conclusion, timely degradation of PCM1 by DNAJA2/HSC70-mediated CMA is essential for maintaining mitotic integrity.

**Defects in DNAJA2/HSC70-mediated CMA induce micronuclei and activate the cGAS-STING pathway**

Since MN can form upon mitotic exit from anaphase lagging chromosomes or unsegregated chromosome fragments[40], which are present in *DNAJA2*-depleted cells (Fig. 1c and Supplementary Movie 2), we speculate that *DNAJA2*-deficiency induces MN formation in interphase cells. We therefore analyzed interphase cells for MN formation using PicoGreen and DAPI staining. We indeed observed that a significantly higher rate of *DNAJA2*-depleted cells displayed MN in various cell lines,

as compared with control cells (Fig. 6a, b and Supplementary Fig. 4a–c).

If MN formation in *DNAJA2*-depleted cells is due to deregulation of centrosome homeostasis caused by untimely degradation of PCM1 via the CMA pathway, cells that have lost DNAJA2 partner HSC70 or the CMA receptor LAMP2A should exhibit similar phenomena observed in *DNAJA2*-depleted cells. To test this hypothesis, we analyzed the mitotic phenotypes and MN formation in *L2A⁻/⁻* HeLa cells. Like *DJ2⁻/⁻* HeLa cells, *L2A⁻/⁻* cells showed delayed mitotic exit, increased mitotic errors (Supplementary Fig. 5a–c), abnormal spindle morphology (Supplementary Fig. 5d, e), as well as an elevated rate of MN formation (Supplementary Fig. 4d, e). Similar results were also obtained when HSC70's activity was blocked by HSC70 inhibitors VER-155008 (VER) and Apoptozole (Supplementary Fig. 4f, g). Taken together, defects in DNAJA2/HSC70-mediated CMA induce mitotic errors that lead to MN formation.

The formation of MN can activate the cytosolic DNA sensor cGAS[41], which in turn activates the STING-mediated type I interferons[42]. We therefore tested activation of the cGAS-STING signaling pathway in *DNAJA2*-deficient cells. Indeed, we observed significantly increased levels of phosphorylated STING, TBK1, and STAT1 in *DNAJA2*-depleted

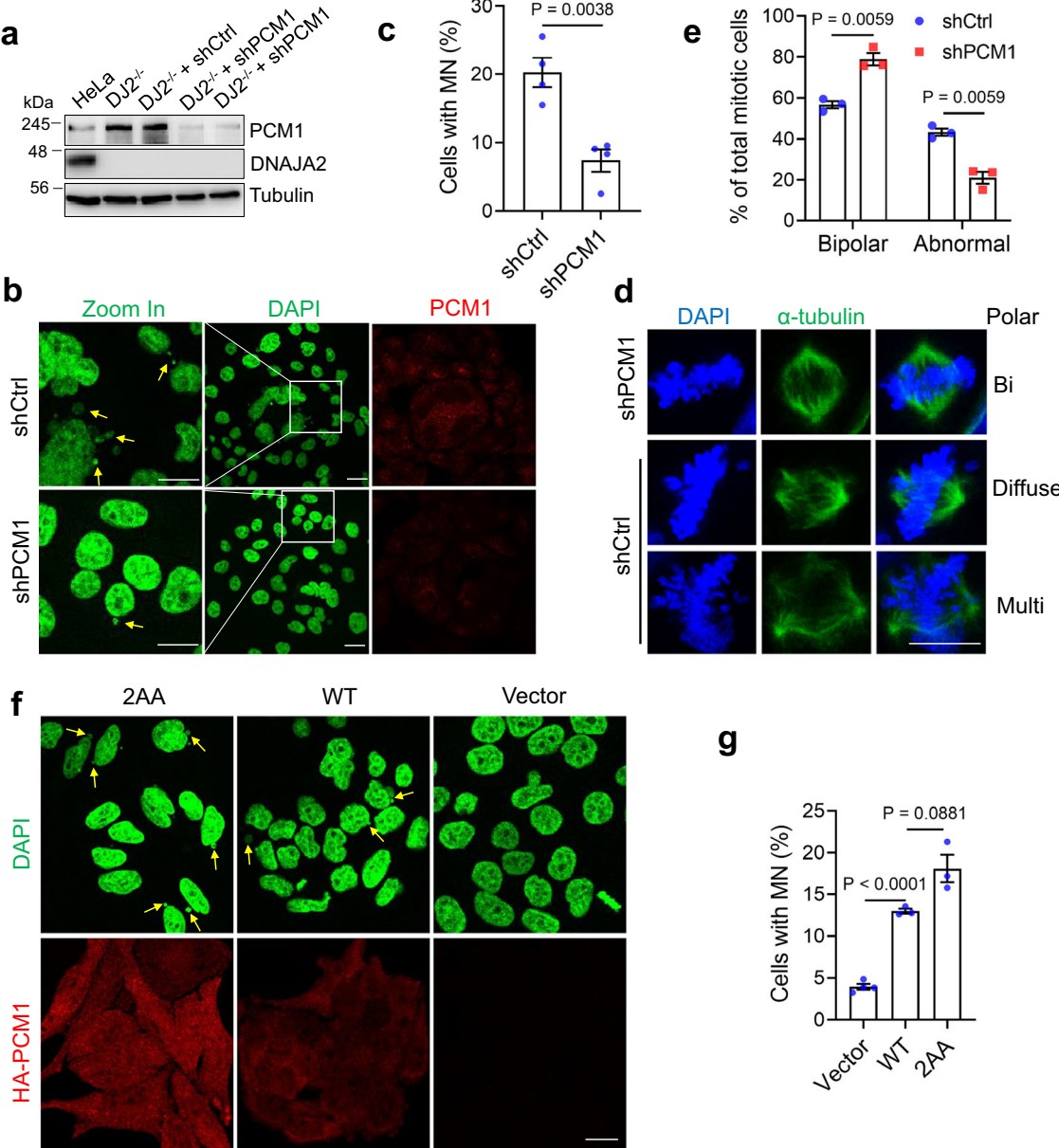

**Fig. 5 | Elevated PCM1 level contributes to the mitotic defects in DNAJA2-deficient cells. a** Western blots showing PCM1 expression levels in WT, *DJ2*$^{-/-}$ and *DJ2*$^{-/-}$ with control shRNA (DJ2$^{-/-}$+shCtrl) or shRNA targeting *PCM1* (DJ2$^{-/-}$+shPCM1) HeLa cells. **b, c** Representative images and quantification (means ± SEM, *n* = 4 experimental repeats) of cells with micronuclei in DJ2$^{-/-}$+shCtrl and DJ2$^{-/-}$+shPCM1 HeLa cells. Micronucleus were indicated by yellow arrows. **d** Representative images showing spindle morphologies (green) of DJ2$^{-/-}$+shCtrl and DJ2$^{-/-}$+shPCM1 HeLa cells. **e** Quantification (means ± SEM, *n* = 3 experimental repeats) of cells with bipolar and abnormal spindles in DJ2$^{-/-}$+shCtrl and DJ2$^{-/-}$+shPCM1 HeLa cells. All multipolar, monopolar, and diffuse-polar spindles were counted as abnormal spindles. **f, g** Representative images and quantification (means ± SEM) of cells with micronuclei in HeLa cells expressing empty vector (*n* = 4 experimental repeats), WT (*n* = 3 experimental repeats), and KFERQ mutant (2AA) (*n* = 3 experimental repeats) HA-PCM1. Scale bar, 20 µm. *P* values were determined by two-tailed unpaired t test with Welch's correction. Source data are provided as a Source data file.

4T1, B16-OVA, H460, and SW620 cells, but the increased phosphorylation disappeared when DNAJA2 expression was restored in the knockout cells (Fig. 6c and Supplementary Fig. 6a, b). Consistently, *DNAJA2*-depleted cells displayed higher expression levels of type I interferon *IFNβ*, interferon-stimulated gene (ISG) *ISG15*, interferon-regulatory factor *IRF7* and C-X-C motif chemokine ligand 10 (CXCL10) (Fig. 6d, e and Supplementary Fig. 6c, d). To validate whether the activation of type I interferon depends on cGAS-mediated DNA sensing, we generated *DNAJA2* and *cGAS* double knockout (DKO) 4T1 cell line. As anticipated, the levels of phosphorylated STING, TBK1, STAT1, and ISG expression significantly reduced in the double knockout cells, as compared with *DNAJA2*-depleted cells (Fig. 6e–g).

Similarly, phosphorylation of IRF3 and STAT1 were also abolished in *DNAJA2* and *STING* double knockout B16-OVA cells (Supplementary Fig. 6e). Collectively, these data demonstrate that *DNAJA2*-deficiency activates the cGAS-STING pathway, leading to the type I interferon production.

Similar to *DNAJA2*-depleted cells, *L2A*$^{-/-}$ 4T1 cells also displayed higher levels of phosphorylated STING, TBK1, and STAT1 (Fig. 6h), as well as higher expression levels of *IFNβ*, *IRF7*, and CXCL10 (Fig. 6e, i). The elevated STAT1 phosphorylation was also observed in *DNAJA2*- and *LAMP2A*-depleted HeLa cells (Supplementary Fig. 6f). To determine if the observed cGAS-STING pathway activation in *DNAJA2*-deficient or *LAMP2A*-deficient cells is related to the role of HSC70/DNAJA2 in

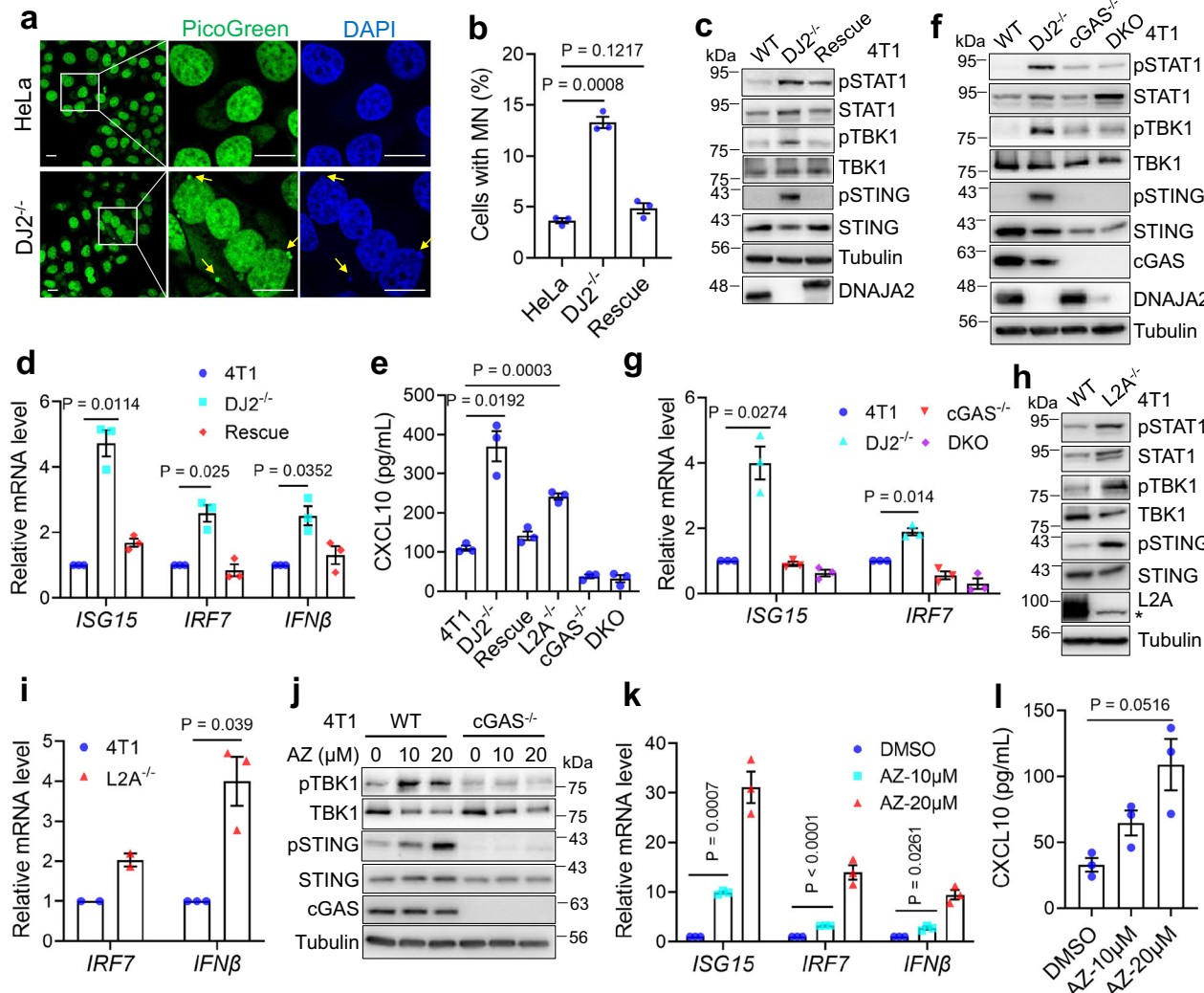

**Fig. 6 | Defects in DNAJA2 or CMA induce micronuclei to activate the cGAS-STING pathway. a**, **b** Representative images and quantification (*n* = 3) of cells with micronuclei in control and *DJ2⁻/⁻* HeLa cells. Micronucleus (see yellow arrows) were stained by PicoGreen (green) and DAPI (blue). Scale bar, 20 μm. **c** Western blots showing elevated phosphorylation of STAT1 (pSTAT1), TBK1 (pTBK1), and STING (pSTING) in *DJ2⁻/⁻* 4T1 cells, *n* = 4 independent experiments. **d** RT-qPCR analysis (*n* = 3) showing relative mRNA levels of *ISG15*, *IRF7*, and *IFNβ* in WT, *DJ2⁻/⁻*, and *DJ2*-rescued 4T1 cells. **e** Elisa assay showing CXCL10 level (*n* = 3) in culture media from various isogenic 4T1 cell lines cultured for 4 days as indicated. **f** Western blots showing cGAS-STING pathway activation in *DJ2⁻/⁻* 4T1 cells, but not in WT, *cGAS⁻/⁻* and *DNAJA2/cGAS* double knockout (DKO) 4T1 cells, *n* = 3 independent experiments. **g** RT-qPCR analysis (*n* = 3) showing relative mRNA levels of *ISG15* and *IRF7* in WT, *DJ2⁻/⁻*, *cGAS⁻/⁻*, and *DNAJA2/cGAS* double knockout (DKO) 4T1 cells. **h** Western

blots showing expression levels of phosphorylated STING, TBK1 and STAT1 in WT and *L2A⁻/⁻* 4T1 cells. * indicates a non-specific band, *n* = 4 independent experiments. **i** RT-qPCR analysis of relative mRNA levels of *IRF7* (*n* = 2) and *IFNβ* (*n* = 3) in WT and *L2A⁻/⁻* 4T1 cells. **j** Expression levels of phosphorylated STING and TBK1 in WT and *cGAS⁻/⁻* 4T1 cells treated with 10 μM or 20 μM Apoptozole (AZ) for 24 h, as indicated, *n* = 3 independent experiments. **k** RT-qPCR analysis (*n* = 3) of relative mRNA levels of *ISG15*, *IRF7*, and *IFNβ* genes in 4T1 cells treated with 10 μM or 20 μM AZ, as indicated. **l** Elisa assay showing CXCL10 level (*n* = 3) in culture media from 4T1 cells treated with 10 μM or 20 μM AZ. Cells were grown for 3 days in total. Data are shown as means ± SEM of *n* experimental repeats. *P* values were determined by two-tailed unpaired t test with Welch's correction. Source data are provided as a Source data file.

centrosome homeostasis and mitotic integrity, we first analyzed the innate immune signaling after treating 4T1 and B16-OVA cells with HSC70 inhibitors Apoptozole (AZ) or VER-155008 (VER). Inhibition of the HSC70 activity dramatically upregulated the levels of phosphorylated STING and TBK1 (Fig. 6j and Supplementary Fig. 6g, h), the type I interferon and ISGs (Fig. 6k, l), which appears to be dependent on cGAS (Fig. 6j). We then analyzed the phosphorylation of TBK1 and STAT1 in PCM1 partial knockdown *DJ2⁻/⁻* HeLa cells. As shown in Supplementary Fig. 6i, PCM1 knockdown significantly reduces the level of phosphorylated TBK1 and STAT1. We therefore conclude that deficiency in the DNAJA2/HSC70-CMA axis causes aberrant mitosis and chromosome instability that activates the cGAS-STING signaling pathway.

## DNAJA2-deficiency enhances immune-checkpoint blockade therapy

Activation of type I interferon by the cGAS-STING signaling promotes immune-checkpoint blockade (ICB) therapy[43–45]. To evaluate if *DNAJA2*-deficiency enhances ICB efficacy, we injected WT or *DNAJA2*-depleted 4T1 or B16-OVA cells subcutaneously into immunocompetent mice, and monitored the tumor growth after treating tumors with or without ICB antibodies. As shown in Fig. 7a, b, the growth of *DNAJA2*-deficient 4T1 or B16-OVA tumors was dramatically inhibited by the ICB treatment, however, WT tumors only showed very limited response to the treatment. To validate if the therapeutic potency of *DNAJA2*-deficient tumors depends on the type I interferon pathway, we inoculated *DNAJA2*-deficient 4T1 cells in WT BALB/c mice, and treated tumors with

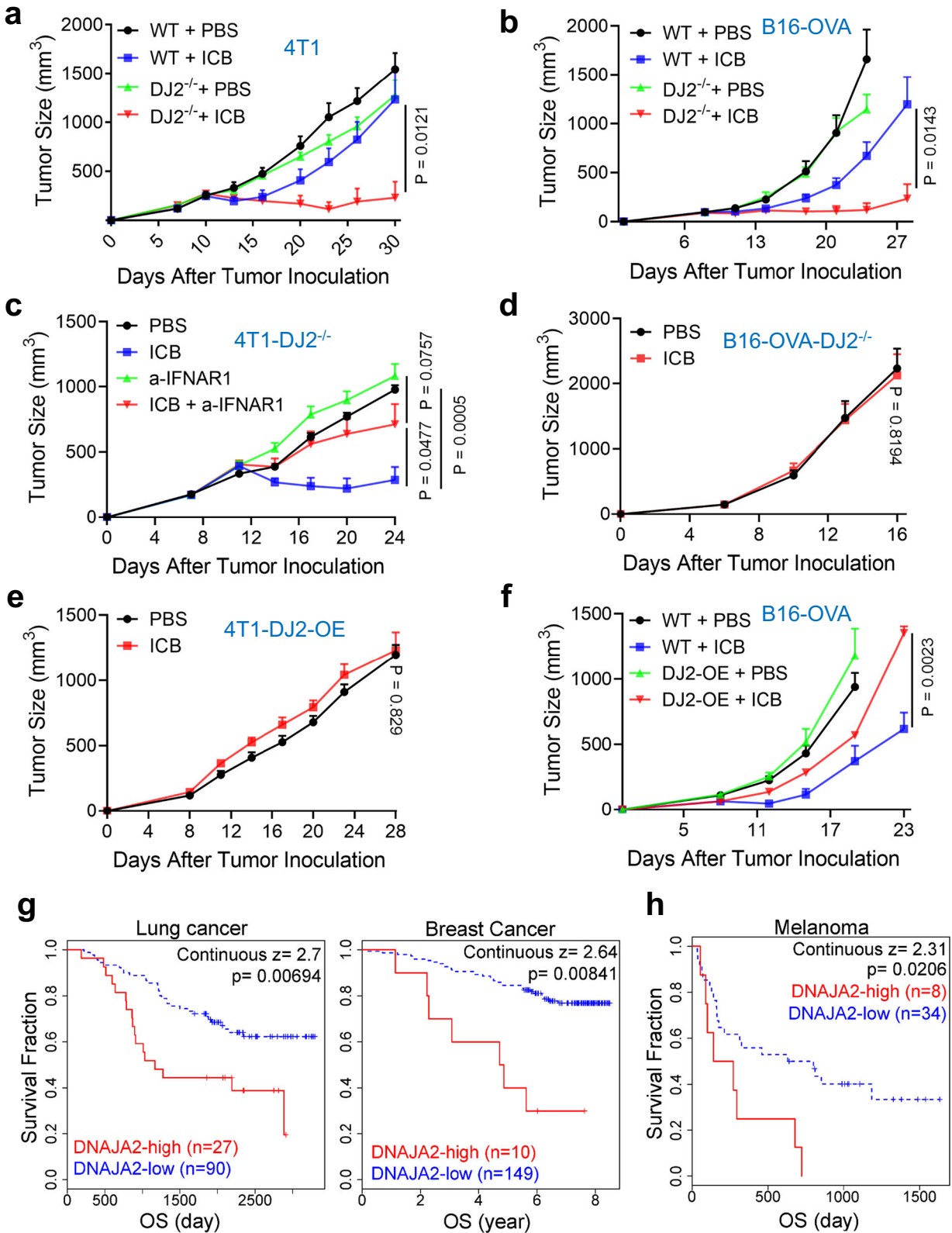

ICB antibodies or anti-IFNAR1 antibody alone, or combination of these antibodies. The results showed that although the growth of *DNAJA2*-deficient tumors was significantly inhibited by ICB treatment alone, the combination treatment abolished the therapeutic potency completely (Fig. 7c). Similarly, when *DNAJA2*-deficient B16-OVA cells were inoculated in *IFNAR1* knockout mice, the potency of ICB treatment was also completely diminished (Fig. 7d). Taken together, these observations

suggest that *DNAJA2*-deficiency facilitates ICB therapy in a type I interferon-dependent manner.

**Upregulations of DNAJA2 and CMA factors contribute to ICB therapy resistance**

To determine the clinical relevance of our findings, we first analyzed *DNAJA2* mRNA levels in various human tumors, and found that

**Fig. 7 | DNAJA2 expression level modulates ICB therapy efficacy. a, b** Tumor growth curves of WT and *DJ2⁻/⁻* tumor cells treated with or without ICB therapy. WT or *DJ2⁻/⁻* 4T1 tumor cells (**a**) were inoculated into WT BALB/c mice (*n* = 7 mice) and treated with anti-CTLA4 and anti-PD-L1 antibodies at day 7, 10, and 13. B16-OVA tumor cells (**b**) were inoculated into WT C57BL/6 mice (*n* = 7 mice) and treated with anti-PD-L1 antibody at day 9, 12, and 15. **c** Tumor growth curves of *DJ2⁻/⁻* 4T1 cells treated with ICB (anti-CTLA4 and anti-PD-L1), anti-IFNAR1 or both. *DJ2⁻/⁻* 4T1 cells were inoculated in WT BALB/c mice (*n* = 6 mice) and treated with ICB antibodies and anti-IFNAR1 mAb at day 8, 11, and 14. **d** Tumor growth curves of *DJ2⁻/⁻* B16-OVA cells inoculated in IFNAR1-KO C57BL/6 mice (*n* = 7 mice). Anti-PD-L1 antibody was administered at day 7, 10, and 13. **e** Tumor growth curves of *DJ2*-OE 4T1 tumors treated with (*n* = 5 mice) or without (*n* = 11 mice) ICB therapy. *DJ2*-OE 4T1 cells were

inoculated in WT BALB/c mice and treated with anti-CTLA4 and anti-PD-L1 antibodies at day 7, 10, and 13. **f** Tumor growth curves of WT and *DJ2*-OE B16-OVA tumor cells treated with (*n* = 5 mice) or without (*n* = 7 mice) ICB therapy. WT or *DJ2*-OE B16-OVA cells were inoculated in C57BL/6 mice and treated with anti-PD-L1 antibody at day 9, 12, and 15. **g** Survival curves of lung and breast cancer patients with low or high expression levels of *DNAJA2*[48]. **h** Overall survival (OS) rate of melanoma patients with high or low expression levels of *DNAJA2* in a clinical cohort treated with anti-CTLA4 therapy[48]. Data are shown as means ± SEM. *P* values in figures (**g**) and (**h**) were determined by Kaplan–Meier analysis, and the others were determined by two-tailed unpaired t test with Welch's correction. Source data are provided as a Source data file.

aberrant expression of *DNAJA2*, including both upregulation and downregulation, is common among the tumors analyzed[46,47] (Supplementary Fig. 7a). We have already demonstrated that DNAJA2-deficient tumors show favorable responses to ICB therapy (Fig. 7a, b). To explore the impact of *DNAJA2* upregulation on ICB treatment, we generated 4T1 and B16-OVA cells with or without *DNAJA2* overexpression (*DJ2*-OE), and injected these cells into immunocompetent mice, followed by ICB treatment. Although the growth of control 4T1 tumors was partially inhibited by ICB treatment (Fig. 7a), tumors with *DJ2*-OE no longer benefited from the treatment (Fig. 7e). More strikingly, the *DJ2*-OE B16-OVA tumors grew much faster than the control tumors after treating with ICB (Fig. 7f), indicating that *DNAJA2* overexpression renders tumor resistance to ICB therapy. Consistent with these results, analysis of cancer database[48] revealed that cancer patients with a lower level of *DNAJA2* (DNAJA2-low) displayed a better surviving rate than those with a higher level of *DNAJA2* (DNAJA2-high) in several types of cancers analyzed, including lung and breast cancers (Fig. 7g), as well as colorectal cancers, neuroblastoma and myeloma (Supplementary Fig. 7b). More strikingly, the data from a melanoma cohort treated with anti-CTLA4 antibody[48,49] showed that patients with lower levels of *DNAJA2* expression were greatly benefitted from the therapy than those with higher levels of *DNAJA2* expression (Fig. 7h). Altogether, these results support that DNAJA2 overexpression confers ICB therapy resistance.

Since DNAJA2 functions via HSC70-mediated CMA, we hypothesized that HSC70 and CMA may impact ICB therapy in a similar manner. To test this possibility, we first analyzed the expression level of *LAMP2* gene in tumors. Unlike *DNAJA2*, *LAMP2* shows dominant upregulation pattern in tumors[46,47] (Supplementary Fig. 7c). To determine if LAMP2A upregulation promotes tumor growth or immune evasion, we injected *LAMP2A*-overexpressing (*L2A*-OE) and control 4T1 cells into immunocompetent mice, and monitored tumor growth after ICB treatment. As shown in Supplementary Fig. 7d, while the growth of control tumors was partially inhibited by ICB treatment, *L2A*-OE tumors grew faster than the control ones and did not respond to ICB treatment. In contrast, low levels of *LAMP2* expression benefit immunotherapy, as patients with a low level of *LAMP2* responded better to the ICB treatment than those with a high level of *LAMP2* (Supplementary Fig. 7e) in an anti-PD1-treated melanoma cohort[48,50]. These results suggest that higher levels of *LAMP2* causes tumor evasion from immunotherapy, but lower levels of *LAMP2* facilitate immunotherapy. Similar conclusion was also drawn for HSC70 after analyzing the expression levels of *HSPA8* (coding HSC70) in patients in a melanoma cohort[48,51] and their response to anti-PD-1 immunotherapy (Supplementary Fig. 7f). We therefore conclude that DNAJA2 and CMA components are promising biomarkers and/or targets for enhancing cancer immunotherapy.

## Discussion

ICB therapy is a great breakthrough discovery in cancer treatment. However, only can a limited fraction of patients benefit from it because of both intrinsic and adaptive resistance mechanisms[52]. Here, we show

that the expression levels of DNAJA2 and CMA factors modulate ICB potency by regulating the type I interferon signaling in tumor microenvironment. Our findings provide a strategy for cancer therapy by specifically targeting on DNAJA2, which may advance HSPs-based cancer therapy because of HSP40's better specificity[4,12,13]. Although dysfunction of CMA has similar effect, we believe that targeting DNAJA2 is more feasible than inhibiting the CMA pathway, as the latter has a broad client pool and will likely cause cellular toxicity and therapeutic resistance.

Both DNAJA2 and CMA confer cancer progression and are required for tumor growth[5,6,30,53,54], but the underlying mechanism remains unclear. We provide strong evidence showing that the DNAJA2-mediated CMA regulates mitotic integrity and innate immunity, suggesting the possibilities that tumor cells rely on the DNAJA2-CMA pathway to avoid post-mitotic cell death[55], innate immunity-induced immunogenic cell death (ICD), or immune surveillance. At the molecular level, we show that DNAJA2-mediated CMA is essential for maintaining the centrosome homeostasis, which is critical to genomic integrity[32,33,56–59]. DNAJA2 directly regulates centrosome homeostasis by mediating timely degradation of key centriolar satellite (CS) proteins, including PCM1, via the CMA pathway (Fig. 8). Under normal circumstance, HSC70 recognizes the KFERQ-like motifs of PCM1 and transfers them to lysosome to interact with LAMP2A before undergoing degradation via LAMP2A-mediated translocation. The initial substrate recognition and subsequent transportation by HSC70 are facilitated by DNAJA2, which interacts with both HSC70 and PCM1. The coordination between DNAJA2, HSC70, and PCM1 results in timely degradation of PCM1 through CMA (Fig. 8a), which maintains the homeostasis of centriolar satellite. This ensures centrosome organization, bipolar spindle formation, and accurate chromosome segregation (Fig. 8b). However, in DNAJA2-deficient cells, centriolar satellite proteins are upregulated. It is these upregulated centriolar satellite proteins that assemble aberrant centrosomes to promote multipolar or diffused spindle formation, leading to chromosome missegregation (Figs. 1 and 5). These chromosome segregation errors cause aberrant mitotic divisions, resulting in daughter cells with multinuclei and micronuclei, which activates the cGAS-STING pathway (Fig. 8c).

Interestingly, previous studies have demonstrated that two other proteolysis pathways, the ubiquitination proteasome system (UPS) and macroautophagy, can also degrade PCM1 to maintain centrosome homeostasis[32,34–36]. Although the reason why a single protein is regulated by multiple pathways is unknown, it may be related to tissue- or cell-type specificity. For example, we showed that lysosome inhibitor CQ but not proteasome inhibitor MG-132 can stabilize PCM1 in 4T1 cells under our experimental conditions, implying that the UPS pathway may play a less important role in regulating PCM1 homeostasis in 4T1 cells. Moreover, this may be also related to the multiple functions of both PCM1 and centrosome in cell metabolism. As a key centriolar satellite protein, PCM1 regulates centrosome assembly and microtubule organization[31]. In addition, PCM1 regulates both UPS and macroautophagy pathways[34,60]. Similarly, centrosome functions

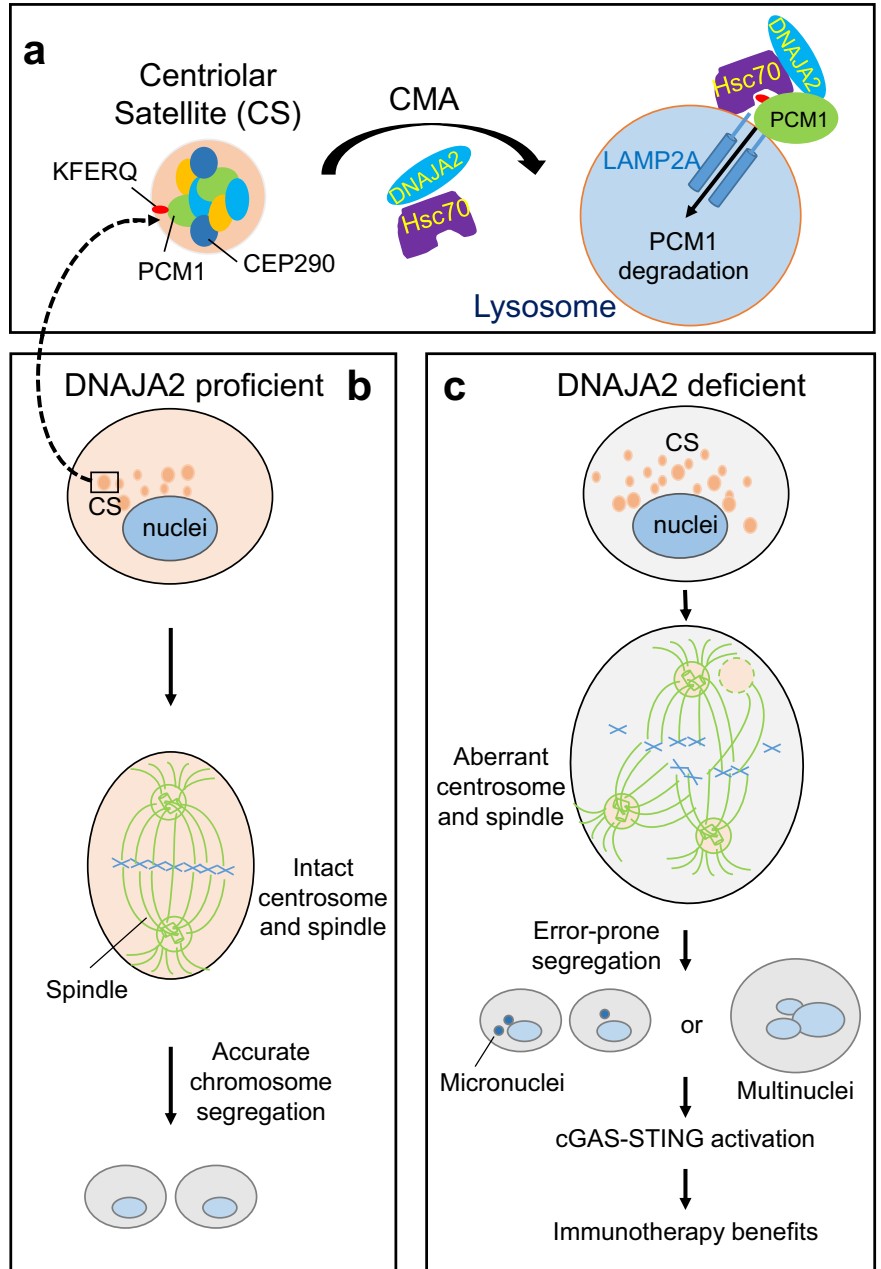

**Fig. 8 | Proposed mechanism by which DNAJA2 regulates mitosis and immunotherapy. a** DNAJA2 facilitates PCM1/CEP290 degradation by HSC70-mediated CMA. PCM1 and CEP290 are important components of centriolar satellites (CS), which assemble and organize centrosomes. To maintain the homeostasis of PCM1, the HSC70 chaperone, together with DNALA2, interacts with PCM1 by recognizing the PCM1 KFERQ-like motifs and transports the latter to lysosomes for degradation via the CMA pathway in a LAMP2A-dependent manner. The timely degradation of PCM1 ensures correct assemble of centrosomes and promotes bipolar spindle formation and accurate chromosome segregation (**b**). However, in the absence of DNAJA2, cells accumulate elevated CS proteins, which results in centrosome/spindle pole fragmentation, abnormal spindles and chromosome missegregation, leading to mitotic failure (multinuclear cells) and micronuclei formation (**c**). This activates the cGAS-STING signaling to promote immune-checkpoint blockade therapy.

beyond mitotic division[61]. Emerging evidence shows that centrosome contains proteins of proteolysis pathways and serves as a center to maintain cellular proteome homeostasis[60,62–64]. It is possible that each of the individual proteolysis systems is responsible for degrading PCM1 and other centrosome components involved in a given function/location, and that the specific recognition target for these substrate proteins could be a particular domain and/or post-translational modification. For example, PCM1 is phosphorylated by a number of protein kinases on multiple residues such as Ser372[39], Thr703, and Ser110[38]. PCM1 can also be mono- and poly-ubiquitinated[34]. These modifications modulate PCM1's properties and location[65]. Here we showed that

inhibiting PCM1 phosphorylation on Ser110 and Ser372 by PLK1 and PLK4, respectively, blocks PCM1 degradation via lysosomal pathway. Since these serine residues, especially Ser110, is very close to the $^{101}$EKLKQ$^{105}$ motif, we believe that the DNAJA2/HSC70-involved CMA functions to degrade phosphorylated PCM1 and regulate PCM1's role in mitotic division. However, thorough investigations are required to verify these possibilities.

CMA mainly functions in protein quality control and selective proteome remodeling in cells, thus playing important roles in diverse physiological functions, to maintain cellular homeostasis[30], including cell cycle regulation[66–68] and DNA damage response[37]. Here, we have

demonstrated a CMA function in maintaining mitotic integrity and cell division by regulating centrosome homeostasis. Since chromosome instability (CIN) derived from defective mitosis promotes cellular transformation and tumorigenesis, our observations here have implicated CMA as an important pathway for mitotic maintenance and tumor suppression[30]. Previous studies have shown that LAMP2A is overexpressed in various tumors, and CMA is required for tumor cell growth, suggestive of the tumor-promoting role of CMA[30,53,54]. Our data that CMA is essential for maintaining tumor cell mitotic division provides further explanation for this, as tumor cells showing CIN phenotype are particularly sensitive to mitosis disruption, which causes chromosome catastrophe and mitotic cell death[55]. In addition, we revealed that LAMP2A overexpression renders tumor resistance to the ICB therapy, implying that CMA may also promote tumor cell growth through immune evasion. Future studies are required to fully understand the roles of CMA in tumor development and control.

In conclusion, this study uncovers a role of the DNAJA2-CMA axis in maintaining genome stability in human cells and provides strategies to advance the HSPs-based cancer therapy by targeting HSP40s in cancer immunotherapy.

## Methods

All mouse experiments were conducted according to regulations of the Institutional Animal Care and Use Committee of the University of Texas Southwestern Medical Center.

### Antibodies, chemicals, and other reagents

All the antibodies, chemicals, oligonucleotides, and plasmids used in this study are listed in Supplementary Data 1.

**Constructs and shRNAs.** All the oligonucleotides used for cDNA and shRNA construction are listed in Supplementary Data 1. Plasmids expressing *DNAJA2* and *LAMP2* were constructed using regular ligation assay using T4 DNA ligase. Plasmids expressing mutant *PCM1* were constructed using the NEBuilder HiFi DNA Assembly Master Mix (Cat# E2621). Oligonucleotides for shRNA construction were annealed and ligated into pLKO.1 vector using regular ligation assay.

### Cell lines, cell culture, and chemical treatment

Human cell lines HeLa, hTERT-RPE1 (kindly provided by Dr. Hongtao Yu), H460, SW620 and mouse breast cancer cell line 4T1 as well as melanoma cell B16-OVA were used in this study. All cells were routinely tested for mycoplasma contamination. Unless mentioned otherwise, all cells were cultured in 37 °C incubator supplemented with 5% $CO_2$; all knockout cell lines were generated using CRISPR-Cas9 technologies[69]. HeLa cell was grown in RPMI 1640 media supplemented with 10% FBS. hTERT-RPE1, H460, SW620, 4T1 and B16-OVA cells were cultured in Dulbecco's modified Eagle medium (DMEM) supplemented with 10% FBS. Cells overexpressing *DNAJA2*, *LAMP2A*, WT or 2AA mutant *PCM1* were constructed by transfecting the pLenti plasmids containing the corresponding coding sequences using jetPRIME® Transfection reagent (PolyPlus# 114-07), followed by puromycin selection and single colony verification. The details of chemical treatments are described in the related figure legends.

### Mouse strains

WT C57BL/6J, BALB/c female mice, and B6[Cg]-Ifnar1tm1.2Ees/J [Ifnar1−/−] female mice were purchased from the Jackson Laboratory. All mice were maintained in a specific pathogen-free animal facility with controlled temperature (68−74 degrees Fahrenheit), 65% humidity, light/dark cycle (lights between 6 am and 6 pm). All experiments were conducted according to regulations of the Institutional Animal Care and Use Committee of the University of Texas Southwestern Medical Center.

### Indirect immunofluorescence and quantification of mitotic phenotypes

Cells were cultured on the cover slides and fixed with 4% paraformaldehyde in PBS for 10 min at room temperature, followed by permeabilization in 0.25% Triton X-100 for 10 min. The slides were blocked by 5% BSA in PBS for 30 min and subsequently incubated with primary antibodies and secondary antibodies each for 2 h at room temperature. After final washing with PBS, the slides were mounted with DAPI solution before imaging. To determine colocalization of DNAJA2 and HSC70 with centrosome proteins, cells were pre-extracted in 0.25% Triton X-100 in PBS for 10 min before fixation. To visualize mitotic spindle morphologies and chromosome alignment defects, cells were grown to 80−90% confluence to enrich mitotic cells without any mitosis inhibitor treatment. All images were taken using a Leica TCS SP8 confocal microscope, and analyzed and quantified using the NIH ImageJ 1.48 software. Colocalization analysis was performed with the Colocalization Finder plugin.

### Time-lapse imaging

WT, $DNAJA2^{-/-}$, or $LAMP2A^{-/-}$ HeLa cells were transfected with H2B-GFP and mCherry-α-tubulin plasmids, and cultured for another 24−48 h before imaging. Time-lapse images were acquired at 2 min intervals using a Leica TCS SP8 microscope equipped with a TOKAI HIT stage top incubator system. Images were processed using the LAS X (Leica) 4.1.0 and NIH ImageJ 1.48 software.

**Measurement of lysosomal protein degradation rate.** The measurement of protein degradation rate for PCM1 and CEP290 in lysosomes was performed as described previously with slight modifications[37]. HeLa and 4T1 cells were treated with or without NL (20 mM ammonium chloride and 100 µM leupeptin) for 16 h and 8 h, respectively, before harvesting for Western blotting analysis. Lysosomal degradation was calculated as the percentage of proteins stabilized by the treatment: degradation per hour (%) = 100%(Relative protein level of NL-treated group/Relative protein level of NL-untreated group − 1)/Treatment time.

### Co-immunoprecipitation and Western blots

Cells were incubated with lysis buffer (50 mM Tris-HCl pH 7.4, 150 mM NaCl, 1% (v/v) NP-40, 1 mM EDTA, 1% sodium deoxycholate) supplemented with protease inhibitor cocktail on ice for 30 min. The cell lysates were centrifuged at $18,000 \times g$, 4 °C for 15 min and supernatants were incubated with a primary antibody overnight, and the protein-antibody conjugates were incubated with Pierce™ Protein G Agarose beads (Thermo Scientific™) for 2 h. After extensive washing with lysis buffer containing increased concentrations of NaCl, the beads were resuspended and boiled with loading buffer, and the samples were subjected to SDS-PAGE and Western blot analysis using antibodies against proteins of interest.

For Western blot analysis of whole cell lysates, cells were lysed in lysis buffer described above supplemented with 1% SDS on ice for 30 min. The lysates were subjected to centrifugation and supernatants were boiled for the subsequent SDS-PAGE and Western blot analysis. All blot images were quantified using Image Lab Software (Bio-Rad).

### RNA isolation and quantitative real-time PCR

Total RNAs from various cell lines were isolated using the Trizol reagent (Invitrogen™, #15596026). Reverse transcription was performed using the qScript cDNA Synthesis Kit (Quantabio, #95047). Quantitative real-time PCR was performed with the SsoAdv Univer SYBR GRN SMX (Bio-Rad, #1725272). The primers used are as follows: mouse *ISG15* (Forward: GAGCTAGAGCCTGCAGCAAT; Reverse: TCACGGACACCAGGAAATCG), mouse *IRF7* (Forward: TTGGGCAAGAC TTGTCAGCA; Reverse: ATACCCATGGCTCCAGCTTC), mouse *IFNB1* (Forward: CCAGCTCCAAGAAAGGACGA; Reverse: CGCCCTGTAGGTG

AGGTTGAT) and Mouse *GAPDH* (Forward: CAACTGCTTAGCCCCCC TGG; Reverse: GCAGGGTAAGATAAGAAATG).

**CXCL10 Elisa.** Cell culture media were collected and centrifuged at $18,000 \times g$, 4 °C for 10 min. The supernatants were subjected to Elisa analysis using the IP-10 (CXCL10) Mouse ELISA Kit (Invitrogen, #BMS6018) or IP-10 (CXCL10) Human ELISA Kit (Invitrogen, #KAC2361) according to the manufacture's instructions. For each cell line, at least three independent samples were collected for the analysis.

## Tumor growth and treatments
WT 4T1 and B16-OVA, and their derivative cell lines, were injected subcutaneously into the right flanks of WT BALB/c and C57BL/6 female mice or the indicated genetically engineered mice at $8 \times 10^5$ cells per mouse. Seven to nine days later, ICB (50–200 µg/mouse anti-CTLA4 and 100 µg/mouse anti-PDL1) was administered every 3 days for a total of 3 times. Tumor size was measured twice weekly and calculated by the following formula: Length × Width × Width/2. Mice were euthanized if length or width of tumor exceeded 2 cm in compliance with the animal protocol. This limit was not exceeded. For IFNAR1 blocking experiments, anti-IFNAR1 mAb was injected intraperitoneally at 200 µg/mouse on the same days of ICB administration for a total of 3 times. All experiments were performed in compliance with the UTSW Human Investigation Committee protocol and UTSW Institutional Animal Care and Use.

## Clinical relevance analysis
The mRNA expression data of *DNAJA2* and *LAMP2* genes were extracted from the cBioPortal database (https://cbioportal.org)[46,47]. The overall survival data for *DNAJA2* in cancer patients and the ICB therapy response data in clinical cohorts were extracted from Tumor Immune Dysfunction and Exclusion (TIDE) database (http://tide.dfci.harvard.edu)[48].

## Statistics and reproducibility
Statistical analyses were performed in GraphPad Prism 9.0 using two-tailed unpaired Student's t-tests. All data were shown as means ± SEM, unless otherwise stated, n indicates the number of replicates or independent experiments in figure legends. A value of $P < 0.05$ was considered statistically significant. No statistical method was used to predetermine sample size and no data were excluded from the analyses. For mice experiments, mice were randomized appropriately in terms of age and weight. Randomization was not performed for in vitro experiments.

# Data availability
All data are available in the main text or the supplementary materials. The mRNA data of *DNAJA2* and *LAMP2* genes were extracted from the cBioPortal database (https://cbioportal.org)[46,47]. The overall survival data and ICB therapy response data in cancer patients were extracted from Tumor Immune Dysfunction and Exclusion (TIDE) database (http://tide.dfci.harvard.edu)[48]. Source data are provided with this paper.

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

## Acknowledgements

We thank Drs. Francesco Cecconi, Li-yuan Yu-Lee, and Hongtao Yu for providing reagents. This work was supported by the Cancer Prevention & Research Institute of Texas grant (CPRIT) RR160101 to G.-M.L. G.-M.L. is a CPRIT Scholar and the Reece A. Overcash, Jr. Distinguished Chair for Research on Conlon Cancer.

## Author contributions

Conceptualization: G.M.L.; funding acquisition and supervision: G.M.L., L.G., and Y.-X.F.; experimental performance and analysis: Y.H.

performed all experiments except the animal experiments (Fig. 7), which were conducted by C.L. H.W. created DNAJA2-KO HeLa cells and performed the initial mitotic division experiments; writing: Y.H. wrote the 1st draft of the manuscript, which was modified by G.-M.L., L.G., Y.-X.F., and C.L.

## Competing interests

The authors declare no competing interests.
