## [Peer Review File · Nature Communications]

Reviewers' Comments:

Reviewer #1:

Remarks to the Author:

In this manuscript, the authors evaluate the role of DNAJA2 deficiency in causing aberrant mitosis, chromosome instability, and subsequent cGAS-STING activation. First, they demonstrate that loss of DJ2 increases chromosome segregation error rate in HeLa cells with abnormal spindles. They extend this finding to murine breast cancer 4T1 cells and also show increased levels of centriolar satellite proteins PCM1 and CEP290. Mechanistically, DNAJA2 regulates PCM21/CEP290 degradation via chaperone mediated autophagy, which they demonstrate using CQ and Lamp2a knockout. They then show that DJ2 inactivation generates micronuclei and activates pSTAT1 and pSTING in 4T1 cells. Treatment of 4T1 or B16-OVA models with anti-PD-1/CTLA-4 with DJ2 knockout results in significantly enhanced response, which is impaired by anti-IFNAR treatment or inoculation in IFNAR KO mice. Finally, elevated DNAJA2 levels in patient tumors correlate with impaired response to ICB.

Overall this is a well conducted study. I have several concerns, however, that should be addressed.

1. Nearly all of the human cell line work is pursued in a single cell line (HeLa). Can the authors replicate the core findings of mitotic aberrations, micronuclei, and cGAS-STING activation in an additional set of human cell lines (eg TNBC, to match the 4T1 mouse model).
2. There are several issues with the assessment of cGAS-STING activation in the main Fig 4 (some of which are addressed in extended data and should be moved to main, such as downstream pIRF3 induction). pSTAT1 and pSTING blots are not sufficient to address pathway activation in the main figure. Total STAT1 and total STING are needed at a minimum as controls. And pTBK1 (with total TBK1 control) is a much more accurate marker of STING activation than pSTING, which is a transient species due to its autophagic degradation. In fact, could the accumulation of pSTING be related to impaired autophagy from DNAJA2 deficiency?
3. Related to the above, I have some concerns about Fig 4i. Normally pSTING shows up as a doublet on a total STING blot. The disconnect between seeing such a dramatic increase in pSTING and no doublet is unusual.
4. Reliance on qPCR alone is not sufficient to measure pathway output. While IFN-beta levels in ELISA can be low following tumor cell STING activation, CXCL10 is a sensitive marker and should be measured in conditioned media in several of the key experiments by ELISA. Related to point 1, most of the cGAS-STING output studies are carried out in 4T1. Does DNAJA2 KO increase STING output as measured by CXCL10 in TNBC lines (those with intact cGAS and STING)?

Reviewer #2:

Remarks to the Author:

In the manuscript, "DNAJA2 deficiency activates cGAS-1 STING pathway and promotes cancer immunotherapy by inducing aberrant mitosis and chromosome instability", the authors detail the role of DNAJA2 (a HSP40 family of chaperone) in regulating mitosis and centrosome maintenance by targeting two main centriolar satellite proteins PCM1 and CEP290 via HSC70 for chaperone-mediated autophagy degradation. The experiments validating PCM1 and CEP290 as CMA substrates lack rigor and there are a few caveats in relation to this aspect, which form major part of my critique for this manuscript.

Fig. 3d-e: The authors show protein stability (with CHX) and less protein levels with AR7 (a CMA inducer). However, these data do not necessarily mean that lysosomal degradation by CMA is responsible for elevated levels of PCM1 and CEP290 in DJ(-) cells. Use of lysosomal inhibitors is key here and the authors are required to assess degradation rates of these proposed substrate proteins via such experiments.

Regarding the KFERQ mutants of PCM1: Please mention if the protein sequence shown in extended data is human or mouse. Also, highlight the exact motif and the mutations made (KFERQ and KFEAA is a generic way of labeling). What is the basis for using only 2 of the 5 KFERQ-motifs in the mutation experiments? Furthermore, the authors presumably only consider canonical motifs. PCM1 has several phosphorylation- and acetylation-generated KFERQ-motifs; why are those motifs not considered? Given that PCM1 is known to be degraded by the proteasome and macroautophagy, this PTM-generated motifs may be a way for the cell to divert the same protein to one pathway of degradation or another. The LIR motifs on PCM1 have been described previously. Exploring some of these aspects of 'which form' of PCM1 is targeted for CMA degradation vs. other pathways will add a significant advance to the story.

The authors start off with both CEP290 and PCM1 as potential CMA substrates but then discard CEP290 and focus only on PCM1. What is the reason for excluding CEP290?

Does the expression of 2AA-PCM1 lead to mitosis aberration and/or micronuclei? Fig. 3g-h: If 2AA mutant is degraded less via CMA, shouldn't its protein levels be more comparable to WT? It doesn't seem to be the case by the blots and numbers in Fig. 3h.

None of the figure panels in Fig. 3, the figure that forms the basis of most of the evidence for PCM1 and CEP290 as new CMA substrates, have statistical analyses. Without any information of number of replicates, experimental variability, and statistical analyses it is incredibly difficult to assess the rigor and reproducibility of the work. This is the major issue for almost all of the data regarding CMA.

Extended Data Fig. 2e-f: The images do not look quite convincing. Perhaps performing the IF +/- lysosomal protease inhibitors would be a better read-out here.

Discussion can be expanded. PCM1 not being degraded by the proteasome at least in 4T1 cells should be commented on. There are previous reports of CMA regulating cell cycle and DNA repair; please refer to the pertinent literature.

Reviewer #3:

Remarks to the Author:

In this manuscript "DNAJA2 deficiency activates cGAS-1 STING pathway and promotes cancer immunotherapy by inducing aberrant mitosis and chromosome instability" the authors show that loss of the HSP40 member DNAJA2 improves immune checkpoint blockade therapy via the interferon. At the mechanistic level the author report that loss of DNAJA2 results in aberrant mitosis, abnormal spindles, elevated centriole satellite protein abundance, micronuclei in interphase and activation of the cGAS-STING pathway. Based on these findings the authors propose a mechanistic model in which DNAJA2 regulates the abundance of centriole satellite proteins, and that loss of this regulation leads to aberrant mitosis with micronuclei due to abnormal mitotic spindle formation.

The claims about the improved immune checkpoint blockade response are well supported by the data and biomedically relevant, as far as I can judge (although this is not necessarily my expertise). The mechanistic model which relies on the first part of the manuscript, which deals with the molecular and cellular data, is however, not well supported by the data and often over-interpreted. This aspect of the manuscript would have to be massively improved in order for the manuscript to be at a publishable level. I therefore overall cannot recommend acceptance of this manuscript.

Major points:

1. The authors show that loss of DNAJA2 leads to excessive levels at centrosomes of the centriolar satellite proteins PCM1 and CEP290. What about other centriolar or centrosomal proteins? Is the change in abundance of these two proteins specific for those 2 proteins? In the absence of this information, it is impossible to judge how specific this effect is.

2. the authors imply a causal chain of events, starting with over-abundance of PCM1/CEP290 at centrosomes leading to aberrant spindles that result in chromosome segregation errors and micronuclei in interphase. This model is, however, only built on correlations. The authors never tested the causalities of this model. Does partial depletion of PCM1 and CEP290 rescue the spindle defects and mitotic errors? Do the authors observed by live cell imaging specifically mis-segregating chromosomes in those cells that display spindle defects? In general are those spindle defects corrected before cells enter anaphase, as is often the case (Ganem et al., 2009), or do the cells attempt multipolar cell divisions? Without such essential information, it is not possible to draw any strong conclusion.

3. Even with regard to the in vivo data in mice with immune checkpoint inhibitors, it would be highly re-assuring to test whether partial depletion of the centriolar satellite proteins abolishes the effect of DNAJA2 loss.

4. The authors generally assume that the mitotic errors seen in cells lacking DNAJA2 show spindle defects due to a mitotic error. However, spindle defects can also arise due to cell-cycle defects in the preceding interphase or cytokinesis. For example a deregulation of the centrosome duplication cycle or a failed cytokinesis might lead to aberrant centrosome numbers, which would already be visible in interphase.

5. More generally, are the PCM1 and CEP290 levels specifically elevated during mitosis or generally also on interphase centrosomes. Measuring this would also help to understand whether the observed phenotype is specific for mitosis or of general nature.

6. The manuscript also suffers from statistical weaknesses: all significances are measured with a simple student t-test. This is, however, not the right test to use in case of multiple comparisons. The authors would have to take this in account, for example with an ANOVA test.

7. Many, if not all, of the quantitative immunoblotting experiments appear to have been carried out only once, which puts in doubt the robustness of the conclusions. Are the reported differences significant or not. This is particularly true where only weak differences are observed (e.g. Figure 3d)

We thank all three reviewers for their insightful comments and valuable criticisms on our manuscript, all of which have now been addressed with either additional experiments and/or new text revision. These comments have helped us to significantly improve the manuscript. Below is our point-by-point response to these comments/criticisms. For convenience, our responses are shown in blue, with reviewers' comments in black.

Reviewer #1 - STING, in vivo models (Remarks to the Author):

In this manuscript, the authors evaluate the role of DNAJA2 deficiency in causing aberrant mitosis, chromosome instability, and subsequent cGAS-STING activation. First, they demonstrate that loss of DJ2 increases chromosome segregation error rate in HeLa cells with abnormal spindles. They extend this finding to murine breast cancer 4T1 cells and also show increased levels of centriolar satellite proteins PCM1 and CEP290. Mechanistically, DNAJA2 regulates PCM21/CEP290 degradation via chaperone mediated autophagy, which they demonstrate using CQ and Lamp2a knockout. They then show that DJ2 inactivation generates micronuclei and activates pSTAT1 and pSTING in 4T1 cells. Treatment of 4T1 or B16-OVA models with anti-PD-1/CTLA-4 with DJ2 knockout results in significantly enhance response, which is impaired by anti-IFNAR treatment or inoculation in IFNAR KO mice. Finally, elevated DNAJA2 levels in patient tumors correlate with impaired response to ICB.

Overall this is a well conducted study. I have several concerns, however, that should be addressed.

Thanks for the positive comments to our work.

1. Nearly all of the human cell line work is pursued in a single cell line (HeLa). Can the authors replicate the core findings of mitotic aberrations, micronuclei, and cGAS-STING activation in an additional set of human cell lines (eg TNBC, to match the 4T1 mouse model).

Thanks for the suggestion. We have generated DNAJA2 knockout lines in H460 (human lung cancer) and SW620 (human colon cancer) cells. We show mitotic aberrations in human RPE1, H460 and SW620 cells (Extended Data Fig. 1c-e). We also show elevated micronuclei (Extended Data Fig. 4c) and cGAS-STING activation (Extended Data Fig. 6b,d) in these cell lines. Therefore, we believe that this is a general phenomenon.

2. There are several issues with the assessment of cGAS-STING activation in the main Fig 4 (some of which are addressed in extended data and should be moved to main, such as downstream pIRF3 induction). pSTAT1 and pSTING blots are not sufficient to address pathway activation in the main figure. Total STAT1 and total STING are needed at a minimum as controls. And pTBK1 (with total TBK1 control) is a much more accurate marker of STING activation than pSTING, which is a transient species due to its autophagic degradation. In fact, could the accumulation of pSTING be related to impaired autophagy from DNAJA2 deficiency?

Thanks for the insightful suggestion. We have blotted both total and phosphorylated STING, TBK1 and STAT1 in Fig. 6, and the results are consistent with our conclusion that the cGAS-STING pathway is activated in DNAJA2- and LAMP2A-KO cells.

As we know, STING undergoes autophagic degradation and is also a substrate of chaperone-mediated autophagy (CMA) pathway (Hu et al., Immunity 2016, 45(3):555-569). However,

DNAJA2, an HSP40 member, may only target some specific proteins for CMA-mediated degradation (Faust et al., Nature 2020, 587(7834):489-494; Piette et al., Mol Cell 2021, 81(12):2549-2565.e8; Kaushik & Cuervo., Nat Rev Mol Cell Biol. 2018, 19(6):365-381) without affecting the overall autophagy in cells. In addition, we didn't observed significant increase of total STING level in DNAJA2-KO cells (Fig. 6c,f and Extended Data Fig. 6a). We therefore believe that DNAJA2-deficiency-induced micronuclei is the major factor that activates the cGAS-STING signaling.

3. Related to the above, I have some concerns about Fig 4i. Normally pSTING shows up as a doublet on a total STING blot. The disconnect between seeing such a dramatic increase in pSTING and no doublet is unusual.

Thanks for pointing this out. We indeed sometimes can see a doublet, with the upper band being pretty weak, when the membrane is over-exposed. However, in most of the cases, we could not see it. This may be related to the STING antibody we used, as well as the cell lines tested. However, we have cGAS-KO and STING-KO cells to verify the specificity of antibodies against STING and pSTING (Fig. 6f,j and Extended Data Fig. 6e).

4. Reliance on qPCR alone is not sufficient to measure pathway output. While IFN-beta levels in ELISA can be low following tumor cell STING activation, CXCL10 is a sensitive marker and should be measured in conditioned media in several of the key experiments by ELISA. Related to point 1, most of the cGAS-STING output studies are carried out in 4T1. Does DNAJA2 KO increase STING output as measured by CXCL10 in TNBC lines (those with intact cGAS and STING)?

Thanks for the constructive suggestion. We have performed ELISA analyses of CXCL10 in both 4T1 (Fig. 6e) and human cell lines H460 and SW620 (Extended Data Fig. 6d), and the data shows that DNAJA2-deficiency significantly induces CXCL10 expression.

Reviewer #2 - Chaperone mediated autophagy (Remarks to the Author):

In the manuscript, "DNAJA2 deficiency activates cGAS-1 STING pathway and promotes cancer immunotherapy by inducing aberrant mitosis and chromosome instability", the authors detail the role of DNAJA2 (a HSP40 family of chaperone) in regulating mitosis and centrosome maintenance by targeting two main centriolar satellite proteins PCM1 and CEP290 via HSC70 for chaperone-mediated autophagy degradation. The experiments validating PCM1 and CEP290 as CMA substrates lack rigor and there are a few caveats in relation to this aspect, which form major part of my critique for this manuscript.

Fig. 3d-e: The authors show protein stability (with CHX) and less protein levels with AR7 (a CMA inducer). However, these data do not necessarily mean that lysosomal degradation by CMA is responsible for elevated levels of PCM1 and CEP290 in DJ(-) cells. Use of lysosomal inhibitors is key here and the authors are required to assess degradation rates of these proposed substrate proteins via such experiments.

Thanks for the constructive suggestions. We have performed the suggested experiment. Both PCM1 and CEP290 proteins are significantly accumulated in WT HeLa and 4T1 cells treated

with lysosomal protease inhibitors ammonium chloride and leupeptin (NL) but not in their DNAJA2-KO counterparts (Fig. 3c and Extended Data Fig. 2g,i). Calculation of lysosomal degradation rates of these proteins shows that the degradation rate for each protein is much lower in DNAJA2-KO cells than in WT cells (Fig. 3d and Extended Data Fig. 2h,j). Therefore, these results suggest that DNAJA2 is required for CMA-mediated lysosomal degradation of PCM1 and CEP290.

Regarding the KFERQ mutants of PCM1: Please mention if the protein sequence shown in extended data is human or mouse. Also, highlight the exact motif and the mutations made (KFERQ and KFEAA is a generic way of labeling). What is the basis for using only 2 of the 5 KFERQ-motifs in the mutation experiments? Furthermore, the authors presumably only consider canonical motifs. PCM1 has several phosphorylation- and acetylation-generated KFERQ-motifs; why are those motifs not considered? Given that PCM1 is known to be degraded by the proteasome and macroautophagy, this PTM-generated motifs may be a way for the cell to divert the same protein to one pathway of degradation or another. The LIR motifs on PCM1 have been described previously. Exploring some of these aspects of 'which form' of PCM1 is targeted for CMA degradation vs. other pathways will add a significant advance to the story.

Thanks again for the insightful suggestions. The protein sequence in Extended Data Fig. 3b is from human, and we have labeled all the motif sequences and mutations created.

PCM1 protein contains 5 canonical and 13 phosphorylation- or acetylation-generated KFERQ-motifs. We checked literatures but did not find any reported phosphorylation or acetylation sites located in these phosphorylation- or acetylation-generated motifs. Since one accessible motif is enough for substrate recognition and more motifs are not necessarily a better CMA substrate (Dice, Trends Biochem Sci. 1990, 15(8):305-9; Kaushik & Cuervo, Trends Cell Biol. 2012, 22(8):407-17), we first focused on the 5 canonical motifs. Proteome-wide analysis of the CMA targeting motifs revealed that motifs on the solvent-exposed region and protein surface are more accessible to HSC70 chaperone (Kirchner et al., PLoS Biol. 2019,17(5):e3000301). In addition, the predicted protein structure of PCM1 from database (<https://www.uniprot.org/uniprotkb/Q15154/entry#structure>) suggests that the two motifs we targeted on are on the surface of PCM1. Therefore, we constructed the 2AA mutant in our study. We showed that the 2AA mutant blocks the interaction between PCM1 and the HSC70/DNAJA2 chaperone complex (Fig. 4b); the 2AA mutant is more stable than WT PCM1 (Fig. 4c,d) and has a lysosomal degradation rate lower than that of the WT protein (Fig. 4f,g). Taken together, these results support the involvement of these two motifs in CMA-mediated degradation of PCM1.

In response to this comment, we generated a mutant (5AA) by converting RQs to AAs in all 5 canonical motifs (Extended Data Fig. 3b), and found that the 5AA mutant shows a protein stability similar to that of the 2AA mutant, as they both are similarly more stable than WT PCM1 (Extended Data Fig. 3c,d). Therefore, we believe that motifs 1 and 5 are critical for CMA-mediated degradation of PCM1.

Posttranslational modifications (PTMs), including phosphorylation and ubiquitination, within or outside the KFERQ motif, can make a KFERQ motif-containing protein accessible to HSC70 (Kaushik & Cuervo, Nat Rev Mol Cell Biol. 2018,19(6):365-381), and this may happen to PCM1. In addition to ubiquitination, PCM1 also undergoes phosphorylation during cell cycle, e.g., by PLK1 at Ser-110 in G2/M (Wang et al., J Cell Sci. 2013,126(Pt 6):1355-65) and by PLK4 at Ser-

372 in G1 (Hori et al., EMBO Rep. 2016,17(3):326-37). These two serine residues, especially Ser-110, are very close to the ¹⁰¹EKLKQ¹⁰⁵ motif. To test if these phosphorylation events contribute to PCM1's lysosomal degradation via CMA, we measured the lysosomal degradation rate of PCM1 when PLK1 or PLK4 is inhibited by their specific inhibitors Volasertib or Centrinone, respectively. As shown in Fig. 4h,i, lysosome-mediated PCM1 degradation is dramatically blocked by these inhibitors, supporting the idea that phosphorylated PCM1 is targeted for lysosomal degradation by CMA pathway. We also discussed more about this part in the Discussion section.

The authors start off with both CEP290 and PCM1 as potential CMA substrates but then discard CEP290 and focus only on PCM1. What is the reason for excluding CEP290?

Thanks for raising this question. Our data shows that both proteins are upregulated in DNAJA2-KO and LAMP2A-KO cells (Fig. 2c-i, Fig. 3g-k and Extended Data Fig. 2a), as well as in autophagy- and HSC70-inhibited cells (Fig. 3a,b,e,f and Extended Data Fig. 2e,f). Similar to PCM1, the stability of CEP290 is also sensitive to the treatment of ammonium chloride (NH₄Cl) and leupeptin, in a manner dependent on DNAJA2 (Extended Data Fig. 2g,h). These observations suggest that the homeostasis of both PCM1 and CEP290 in mitotic division is regulated by the same DNAJA2-dependent mechanism. Given that PCM1 is considered a master regulator of centrosome organization (Dammermann & Merdes, J Cell Biol. 2002, 159(2):255-66; Holdgaard et al., Nat Commun. 2019, 10(1):4176; Tollenaere et al., Cell Mol Life Sci. 2015, 72(1):11-23), we decided to focus on PCM1 to investigate the role of DNAJA2 in regulating mitosis in this study.

Does the expression of 2AA-PCM1 lead to mitosis aberration and/or micronuclei? Fig. 3g-h: If 2AA mutant is degraded less via CMA, shouldn't its protein levels be more comparable to WT? It doesn't seem to be the case by the blots and numbers in Fig. 3h.

Thanks for this suggestion and pointing out the issue. We have generated HeLa cells expressing WT or 2AA mutant PCM1 as suggested. Overexpression of WT PCM1 significantly induces micronuclei, and 2AA mutant causes even more striking phenotypes (Fig. 5f,g), suggesting that CMA-mediated timely degradation of PCM1 is essential to maintain mitotic integrity and chromosomal stability.

In the original Fig. 3h, we used cells transiently expressing WT and 2AA PCM1 proteins, and the quantification numbers were derived by comparing the protein band intensity of time 0 with times 1h and 3h (the value of time 0 is regarded as 1), rather than comparing between WT and 2AA groups. To clarify the issue, we established cell lines stably expressing WT and 2AA PCM1s. We indeed observed that the protein levels of mutant PCM1 are higher than that of WT PCM1 (Fig. 4c and Extended Data Fig. 3c).

None of the figure panels in Fig. 3, the figure that forms the basis of most of the evidence for PCM1 and CEP290 as new CMA substrates, have statistical analyses. Without any information of number of replicates, experimental variability, and statistical analyses it is incredibly difficult to assess the rigor and reproducibility of the work. This is the major issue for almost all of the data regarding CMA.

Thanks to the constructive comment, we have performed the statistical analysis and shown the results in Fig. 3-4 and Extended Data Fig. 2-3. Also see figure legends for methodology.

Extended Data Fig. 2e-f: The images do not look quite convincing. Perhaps performing the IF -/+ lysosomal protease inhibitors would be a better read-out here.

Thanks for the constructive suggestion. We have performed the IF in WT and DNAJA2-KO HeLa cells in the presence or absence of NL (ammonium chloride (NH₄Cl) and leupeptin). The data is shown in Extended Data Fig. 2k,l. NL treatment significantly enhances PCM1-LAMP2A colocalization in WT but not in DNAJA2-KO cells, supporting the idea that DNAJA2 is required for targeting PCM1 to lysosomal degradation.

Discussion can be expanded. PCM1 not being degraded by the proteasome at least in 4T1 cells should be commented on. There are previous reports of CMA regulating cell cycle and DNA repair; please refer to the pertinent literature.

As suggested, we have added more discussions on PCM1 degradation and CMA's important roles in cell cycle control (e.g., HIF1 α), DNA damage response (e.g., phosphorylated Chk1) and cancer development.

Reviewer #3 - Chromosome segregation, mitosis (Remarks to the Author):

In this manuscript "DNAJA2 deficiency activates cGAS-1 STING pathway and promotes cancer immunotherapy by inducing aberrant mitosis and chromosome instability" the authors show that loss of the HSP40 member DNAJA2 improves immune checkpoint blockade therapy via the interferon. At the mechanistic level the author report that loss of DNAJA2 results in aberrant mitosis, abnormal spindles, elevated centriole satellite protein abundance, micronuclei in interphase and activation of the cGAS-STING pathway. Based on these findings the authors propose a mechanistic model in which DNAJA2 regulates the abundance of centriole satellite proteins, and that loss of this regulation leads to aberrant mitosis with micronuclei due to abnormal mitotic spindle formation.

The claims about the improved immune checkpoint blockade response are well supported by the data and biomedically relevant, as far as I can judge (although this is not necessarily my expertise). The mechanistic model which relies on the first part of the manuscript, which deals with the molecular and cellular data, is however, not well supported by the data and often over-interpreted. This aspect of the manuscript would have to be massively improved in order for the manuscript to be at a publishable level. I therefore overall cannot recommend acceptance of this manuscript.

Thanks for the positive comments on the therapy data in mouse models.

Major points:

1. The authors show that loss of DNAJA2 leads to excessive levels at centrosomes of the centriolar satellite proteins PCM1 and CEP290. What about other centriolar or centrosomal proteins? Is the change in abundance of these two proteins specific for those 2 proteins? In the absence of this information, it is impossible to judge how specific this effect is.

Thanks for the nice suggestion. We have checked the expression levels of other important centrosomal and centriolar proteins, including CEP131, SSX2IP, Pericentrin and Centrin. These proteins don't change much (Extended Data Fig. 2c,d), indicating that the effects are specific for PCM1 and CEP290. This is consistent with the concept that HSP40 proteins determine the

substrate diversity and specificity of HSC70 and each HSP40 member only targets certain number of substrates (Faust et al., *Nature* 2020, 587(7834):489-494; Piette et al., *Mol Cell* 2021, 81(12):2549-2565.e8; Kaushik & Cuervo., *Nat Rev Mol Cell Biol.* 2018, 19(6):365-381).

2. the authors imply a causal chain of events, starting with over-abundance of PCM1/CEP290 at centrosomes leading to aberrant spindles that result in chromosome segregation errors and micronuclei in interphase. This model is, however, only built on correlations. The authors never tested the causalities of this model. Does partial depletion of PCM1 and CEP290 rescue the spindle defects and mitotic errors? Do the authors observed by live cell imaging specifically mis-segregating chromosomes in those cells that display spindle defects? In general are those spindle defects corrected before cells enter anaphase, as is often the case (Ganem et al., 2009), or do the cells attempt multipolar cell divisions? Without such essential information, it is not possible to draw any strong conclusion.

Thanks for the constructive suggestions. Since PCM1 is a master regulator of centrosome organization and plays important roles in mitosis (Dammermann & Merdes, *J Cell Biol.* 2002, 159(2):255-66; Holdgaard et al., *Nat Commun.* 2019, 10(1):4176; Tollenaere et al., *Cell Mol Life Sci.* 2015, 72(1):11-23), we partially knockdown PCM1 in DNAJA2-KO HeLa cells (Fig. 5a), as suggested, and quantified mitotic phenotypes. As shown in Fig. 5a-e, partial knockdown of PCM1 in DNAJA2-KO cells rescues the mitotic phenotypes; however, overexpression of PCM1, especially the CMA-inaccessible mutant (2AA mutant) sufficiently induces high levels of micronuclei (Fig. 5f,g), suggesting that an elevated PCM1 protein level contributes to aberrant mitosis in DNAJA2-deficient cells.

As suggested, we also performed live-cell imaging assays to monitor mitotic spindle (mCherry-tubulin) and chromosomes (H2B-GFP) simultaneously in WT and DNAJA2-KO HeLa cells. The data shows that the majority of the mitotic cells with aberrant spindles (multipolar and diffused spindles) before entering anaphase indeed display chromosome segregation errors (Fig. 1f,g). A fraction of cells with multipolar spindles undergo multipolar division but fail at the cytokinesis step (Fig. 1f,g), resulting in multinuclear cells usually containing micronuclei, which is consistent with the elevated level of multinuclear cells observed in DNAJA2-KO cells (Fig. 1a,b). The other fraction of cells with multipolar spindles and cells with diffused spindles usually correct their spindles before anaphase onset to form pseudo-bipolar spindles (this usually takes much longer time than cells with normal bipolar spindles), and undergo bipolar segregation, but frequently show lagging chromosomes and/or missegregation (Fig. 1g), consistent with the observations in the literatures the reviewer suggested (Ganem et al., *Nature* 2009, 460(7252):278-82). These lagging chromosomes finally form micronuclei, which is consistent with the elevated level of micronuclei we observed in DNAJA2-KO cells (Fig. 6a,b and Extended Data Fig. 4a-c).

In conclusion, our data support the conclusion that DNAJA2-deficiency causes elevated level of PCM1, which induces abnormal centrosomes and spindles, leading to mitotic errors.

3. Even with regard to the in vivo data in mice with immune checkpoint inhibitors, it would be highly re-assuring to test whether partial depletion of the centriolar satellite proteins abolishes the effect of DNAJA2 loss.

Thanks for the suggestion. We show clearly that the immunotherapy benefits gained from DNAJA2 loss is dependent on the cGAS-STING activated type I interferon induction (Fig. 6,7), which relies on the increased cytosolic DNA (micronuclei) resulted from DNAJA2-deficiency (Fig. 6a,b and Extended Data Fig. 4). We also show that partial knockdown of PCM1 in

DNAJA2-KO cells rescues the mitotic phenotypes (Fig. 5a-e) and significantly reverses the cGAS-STING activation (reduced level of phosphorylated STAT1 and TBK1) in DNAJA2-KO cells (Extended Data Fig. 6i), thereby strongly demonstrates that the immunotherapy benefits gained from DNAJA2 loss comes from the mitosis-associated chromosomal instability.

In addition, the mouse studies show nice relevance to the clinical data (Fig. 7g,h and Extended Data Fig. 7), suggesting that DNAJA2 is a potential biomarker and/or target to improve immunotherapy efficacy. Thus, we only focus on DNAJA2 to investigate its therapeutic effects in mouse models. The experiment suggested here is out of the scope of this study.

4. The authors generally assume that the mitotic errors seen in cells lacking DNAJA2 show spindle defects due to a mitotic error. However, spindle defects can also arise due to cell-cycle defects in the preceding interphase or cytokinesis. For example a deregulation of the centrosome duplication cycle or a failed cytokinesis might lead to aberrant centrosome numbers, which would already be visible in interphase.

Thanks for the comment and we agree with the reviewer. To test this possibility, we first performed live-cell imaging to monitor mitotic spindles and found that the majority of mitotic cells show bipolar spindles at the mitotic onset (Fig. 1f and Movie S3), but undergo centrosome fragmentation and form abnormal spindles rapidly later, therefore ruling out the possibility that the spindle defects in DNAJA2-deficient cells arise from the preceding cell cycle. To further confirm this, we quantified centrosomes in fixed interphase cells. As shown in Extended Data Fig. 1a,b, the majority of interphase cells show two well-defined centrosomes in DNAJA2-KO cells, which is similar to the WT cells. Therefore, we believe that the observed spindle defects in DNAJA2-deficient cells do not arise from the preceding cell cycle.

5. More generally, are the PCM1 and CEP290 levels specifically elevated during mitosis or generally also on interphase centrosomes. Measuring this would also help to understand whether the observed phenotype is specific for mitosis or of general nature.

Thanks for the nice suggestion. We have performed immunofluorescence analysis and quantification of PCM1 and CEP290 signals basically in interphase cells (Fig. 2e,f and Extended Data Fig. 2k), which do show significant increase. Also, we observed colocalization between PCM1 and CMA receptor LAMP2A, especially when the lysosomal proteases were inhibited by ammonium chloride (NH₄Cl) and leupeptin treatment (Extended Data Fig. 2k,l), suggesting that the CMA-mediated degradation happens in interphase cells. Therefore, we believe that the phenomenon is not specific for mitosis, but also for interphase cells.

6. The manuscript also suffers from statistical weaknesses: all significances are measured with a simple student t-test. This is, however, not the right test to use in case of multiple comparisons. The authors would have to take this in account, for example with an ANOVA test.

Thanks for pointing out this issue. We noticed that there were some confusing statistical labels in the original Fig. 4f and Extended Data Fig. 2f, which showed multiple comparisons. Actually, we only performed comparisons between two groups (i.e., WT vs DJ2^{-/-}, DJ2^{-/-} vs cGAS^{-/-} and DJ2^{-/-} vs DKO^{-/-}) using unpaired student t-test. We have corrected these mislabels in the revised Fig. 6.

7. Many, if not all, of the quantitative immunoblotting experiments appear to have been carried out only once, which puts in doubt the robustness of the conclusions. Are the reported

differences significant or not. This is particularly true where only weak differences are observed (e.g. Figure 3d)

Thanks for pointing out this issue. We do have multiple independent replicates for each of the immunoblotting experiments, but only show one representative image in the original figures. We have shown quantification plots, statistical results and number of replicates in the revised figures and figure legends.

Reviewers' Comments:

Reviewer #1:

Remarks to the Author:

The authors have satisfactorily addressed my concerns

Reviewer #2:

Remarks to the Author:

In the revised/appeal manuscript the authors have addressed each of my previous concerns. I have no further comments.

Reviewer #3:

Remarks to the Author:

In this revised manuscript the authors have addressed most of my comments regarding the mitotic phenotype, and they provide key data supporting their original hypothesis. I therefore support publication of this interesting manuscript.

Nevertheless, I would suggest one important minor textual revision. In their revised manuscript the author state that loss of DNAJA2 leads to "centrosome fragmentation". There is however, no direct data supporting this conclusion. The authors did not perform live or fixed cell imaging with centrosome markers, but instead relied mostly on tubulin as a marker of the mitotic spindle. Their data clearly shows aberrant bipolar and multi-polar spindles, and I would leave it at the term of "spindle pole fragmentation", which is more pertinent than "centrosome fragmentation".

Reviewer #1 (Remarks to the Author):

The authors have satisfactorily addressed my concerns

We are happy to hear about this.

Reviewer #2 (Remarks to the Author):

In the revised/appeal manuscript the authors have addressed each of my previous concerns. I have no further comments.

We are happy to hear that the reviewer is satisfied with the revised manuscript.

Reviewer #3 (Remarks to the Author):

In this revised manuscript the authors have addressed most of my comments regarding the mitotic phenotype, and they provide key data supporting their original hypothesis. I therefore support publication of this interesting manuscript.

We are happy to hear that the reviewer supports publication of our manuscript.

Nevertheless, I would suggest one important minor textual revision. In their revised manuscript the author state that loss of DNAJA2 leads to "centrosome fragmentation". There is however, no direct data supporting this conclusion. The authors did not perform live or fixed cell imaging with centrosome markers, but instead relied mostly on tubulin as a marker of the mitotic spindle. Their data clearly shows aberrant bipolar and multi-polar spindles, and I would leave it at the term of "spindle pole fragmentation", which is more pertinent than "centrosome fragmentation".

Thanks for the suggestion, and we have modified the text as suggested.